


**Modeling the effects of litter stoichiometry and soil mineral N**
**availability on soil organic matter formation**
Haicheng Zhang[1], Daniel S. Goll[1], Stefano Manzoni[2,3], Philippe Ciais[1], Bertrand
Guenet[1], Yuanyuan Huang[1]
[1]*Le Laboratoire des Sciences du Climat et de l'Environnement, IPSL-LSCE*
*CEA/CNRS/UVSQ Saclay, 91191, Gif-sur-Yvette, France*
[2]*Department of Physical Geography, Stockholm University, Stockholm, Sweden*
[3]*Bolin Centre for Climate Research, Stockholm, Sweden*
Correspondence: Haicheng Zhang (haicheng.zhang@lsce.ipsl.fr)
Type: primary research

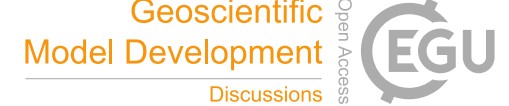

## Abstract

Microbial decomposition of plant litter is a crucial process for the land carbon (C) cycle, as it directly controls the partitioning of litter-C between $CO_2$ released to the atmosphere versus the formation of new soil organic matter (SOM). Land surface models used to study the C cycle rarely considered flexibility in the decomposer C use efficiency ($CUE_d$) defined by the fraction of decomposed litter-C that is retained as SOM (as opposed to be respired). In this study, we adapted a conceptual formulation of $CUE_d$ based on assumption that litter decomposers optimally adjust their $CUE_d$ as a function of litter substrate C to nitrogen (N) stoichiometry to maximize their growth rates. This formulation was incorporated into the widely used CENTURY soil biogeochemical model and evaluated based on data from laboratory litter incubation experiments. Results indicated that the CENTURY model with new $CUE_d$ formulation was able to reproduce differences in respiration rate of litter with contrasting C:N ratios and under different levels of mineral N availability, whereas the default model with fixed $CUE_d$ could not. Using the model with adapted $CUE_d$ formulation, we also illustrated that litter quality affected the long-term SOM formation crucially. Litter with a small C:N ratio tended to form a larger SOM pool than litter with larger C:N ratios, as it could be more efficiently incorporated into SOM by microorganisms. This study provided a simple but effective formulation to quantify the effect of varying litter quality (N content) on SOM formation across temporal scales. Optimality theory appears to be suitable to predict complex processes of litter decomposition into soil C, and to quantify how plant residues and manure can be harnessed to improve soil C sequestration for climate mitigation.

*Keywords*: microbial carbon use efficiency, litter decomposition, litter stoichiometry, soil organic matter, litter decay model, nitrogen





## 1 Introduction

Plant litter decomposition plays a key role in global carbon (C) cycle, thus
needs to be well represented in land surface models. The decomposition and
transformation processes of plant litter control the formation of soil organic matter
(SOM) (Prescott, 2010; Schmidt *et al.*, 2011; Walela *et al.*, 2014; Cotrufo *et al.*, 2015)
and associate immobilization and mineralization of essential plant nutrients
(Moorhead and Sinsabaugh, 2006; Parton *et al.*, 2007; Manzoni *et al.*, 2008; Manzoni
and Porporato, 2009). Hence a reliable litter decay model is necessary for estimating
soil C balance and turnover of ecosystem C (Allison, 2012; Bonan *et al.*, 2013;
Wieder *et al.*, 2013; Campbell and Paustian, 2015). In particular, a realistic
representation of litter decomposition process in land surface models is also helpful to
decrease the uncertainties in predicted effects of climate change and anthropogenic
management on ecosystems (Gholz *et al.*, 2000; Campbell and Paustian, 2015; Luo *et*
*al.*, 2016). As litter decomposition is a very complex process determined by climate
(e.g. temperature and moisture), litter quality (e.g. nitrogen (N) concentration), soil
nutrients and the physiological characteristics of microorganisms (Lekkerkerk *et al.*,
1990; Prescott, 2010; Manzoni *et al.*, 2012; Frey *et al.*, 2013; Sinsabaugh *et al.*, 2013;
Garc á-Palacios *et al.*, 2016), there remain large uncertainties in existing litter decay
models (Zhang *et al.*, 2008; Bonan *et al.*, 2013; Campbell and Paustian, 2015). Many
litter decay models, especially those incorporated in global land surface models, have
ignored microbial mechanisms related to stoichiometry (Bonan *et al.*, 2013; Cotrufo
*et al.*, 2013; Wieder *et al.*, 2013; Wieder *et al.*, 2014).
Microbial carbon use efficiency (CUE), defined as the ratio of microbial
biomass production to material uptake from substrates (Lekkerkerk *et al.*, 1990;
Manzoni *et al*., 2012), is an important emerging property of litter decay, however it
has rarely been represented in land surface models. During litter decomposition, only a
part of the decomposed litter-C is being transferred into SOM, while the remaining C is
being released as $CO_2$ to the atmosphere by microbial respiration. While CUE is a
physiological property of each decomposer community, it also determines the





ecosystem-level efficiency at which litter C is transferred into SOM a step further from
simple microbial incorporation. We denote this efficiency as carbon use efficiency of
litter decomposition ($CUE_d$). With higher $CUE_d$, more plant-produced litter is
transformed biologically into SOM, and soil C storage can reach higher values (Six *et*
*al.*, 2006; Sinsabaugh *et al.*, 2013). In most existing soil biogeochemical models,
$CUE_d$ of decomposition is assumed to be same to microbial CUE and considered as a
fixed parameter. The Verberne model (Verberne *et al.*, 1990) assumes for instance
$CUE_d \approx 0.25$. In the Yasso model (Liski *et al.*, 2005), the $CUE_d$ is set to 0.2.The
CENTURY model (Parton *et al.*, 1988) sets the $CUE_d$ for decomposition of surface
and belowground metabolic litter to 0.55 and 0.45, respectively. In Daisy (Hansen *et*
*al.*, 1991), NCSOIL (Molina *et al.*, 1983) and ICBM (Kätterer and Andrén, 2001),
$CUE_d = 0.6$ for the labile litter pools and takes a lower value for recalcitrant substrates.
Only a few models account for variable $CUE_d$, letting it vary in response to substrate
stoichiometry (Schimel and Weintraub, 2003) or temperature (Allison et al., 2010).

The increasing evidence for a variable microbial CUE leads to a conceptual

CUE model which can explain trends in CUE of microorganisms along stoichiometric
gradients (Manzoni *et al*., 2017). The values of $CUE_d$ used in existing litter decay
models are mostly derived from laboratory study on microbial physiology or limited
observations at some certain ecosystems, thus show large variations (Parton *et al.*,
1988; Verberne *et al.*, 1990; Hansen *et al.*, 1991; Liski *et al.*, 2005; Manzoni *et al.*,
2012). Recent studies (Manzoni *et al.*, 2008, 2012) suggested that the microbial CUE
of terrestrial ecosystems ranges from less than 0.1 for wood decomposers to about 0.5
for decomposition of N-rich and high-quality litter. To explain those differences,
Manzoni *et al*. (2017) proposed a conceptual model of microbial CUE based on the
assumption that decomposers seek to reach an optimum (maximum) growth rate. This
model based on optimality theory links CUE to substrate and decomposers
stoichiometry, where the optimal CUE decreases with increasing substrate
C-to-nutrient ratio, and increases with soil nutrient availability. The predictions of this
theoretical model have been verified by empirical evidence from CUE estimates for
different microorganisms in both aquatic and terrestrial ecosystems (Manzoni *et al.*,

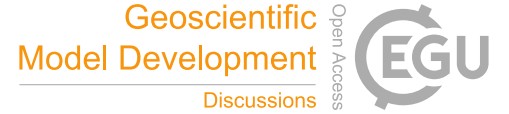



2017).

Besides variable $CUE_d$, many previous studies have also indicated the

necessity for litter decomposition models to consider soil mineral N availability as a
driver of litter decomposition rates, in particular under low N availability (Wieder *et*
*al.*, 2015; Luo *et al.*, 2016; Averill and Waring, 2018). Biomass of microbes is
stoichiometrically constrained (Cleveland and Liptzin, 2007; Franklin *et al.*, 2011;
Allison, 2012). When the supply of N from substrates is lower than the demand of
microbes to fulfill their specific stoichiometric C:N ratio, microbes will utilize the
mineral N (immobilization) (Manzoni *et al.*, 2012). Thus low availability of mineral
N can limit microbial activity, and in turn litter decay rate ( Manzoni and Porporato
2009; Fujita *et al.*, 2014). Although there are fertilization experiments which reported
insignificant or even negative impacts of added N on litter decay rate (Fog, 1988;
Hobbie and Vitousek, 2000; Finn *et al.*, 2015), many incubation experiments showed
a significant decrease of litter decomposition rate with declining mineral N
availability (Recous *et al.*, 1995; Hobbie and Vitousek, 2000; Guenet *et al.*, 2010).
Moreover, recent modeling studies have indicated that the soil biogeochemical model
and Community Land Model could better replicate observed C and N flux when they
included the limiting effect of low mineral N (Bonan *et al.*, 2013; Fujita *et al.*, 2014).
It seems that soil mineral N can alter litter C flux though affecting both the litter
decay rate and the partition of decayed litter-C.

Some detailed microbial decomposition models actually have included

variable microbial CUE and the limitation of low mineral N availability on litter decay
rate (Ingwersen *et al.*, 2008; Pagel *et al.*, 2013; Campbell *et al.*, 2016; Huang *et al.*,
2018), however the parameterization and the evaluating of these models pose
significant challenges due to their complexity and limited verification data (Wieder *et*
*al.*, 2014; Campbell and Paustian, 2015). There is still scope for implementing the
effects of litter stoichiometry and soil mineral N availability on litter decomposition in
litter decay models with more generalizable structure. In particular, it is important to
test the role of these effects in models that have been extensively incorporated into
land surface model for long-term and large-scale application (e.g. CENTURY, Parton



*et al*., 1988). In this study, we incorporated flexible $CUE_d$ based on substrate C:N ratios

and mineral N limitations into a soil biogeochemical model based on the CENTURY

equations to simulate the decomposition and transfer processes of litter-C. The study

was organized as follows. First, the new model was calibrated and tested against data

from laboratory litter incubation experiments for its ability to capture the effect of

variable litter quality and soil mineral N on litter respiration rates (short-term

simulations). Second, the model parameterized assuming flexible $CUE_d$ and mineral

N limitations was used to explore the consequences of such stoichiometric constraints

on the production of soil organic carbon (SOC) (long-term simulations). With these

two modeling analyses, we aimed at linking stoichiometric constraints acting on

short-term (months to years) decomposition dynamics to their consequences on SOC

accumulation occurring at decadal to centennial time scales.

## 2 Materials and methods

2.1 The CENTURY decomposition model

The basis of the litter decay model used in this study is the CENTURY model

(Fig. 1), a first-order decay model that describes decomposition as a function of

substrate availability and quality, clay content, soil moisture and soil temperature

(Parton *et al.*, 1988). Most land surface models (e.g. Kucharik *et al.*, 2000; Sitch *et

al.*, 2003; Krinner *et al.*, 2005) adopted a similar structure to simulate the litter and

soil biogeochemical processes. Dead organic matter in CENTURY is separated into

structural and metabolic litter and three SOM pools (active, slow, passive) with

different turnover times. There is no explicit representation of microbial biomass in

CENTURY, instead the biomass of microbes is assumed to be in equilibrium with

active SOM and thus implicitly included in the active SOM pool. When C is being

transferred between pools, a fraction of it is respired to the atmosphere and the

remaining fraction ($CUE_d$ conceptually equal to microbial CUE) enters the acceptor

pool. Three of such fractions are defined to characterize the transfer of C from the

metabolic litter to the active SOM pool ($CUE_{ma}$), and from the structural litter to





active and slow SOM pool (*CUE_sa* and *CUE_ss*, respectively, Fig. 1). These fractions
are set to be time invariant in the original version of CENTURY, so that a fixed
fraction of decomposed C is retained in the acceptor pool regardless of environmental
conditions and changes in the quality of the donor pool. The N flows in CENTURY
follow the C flows and are equal to the product of C flow by the N:C ratio of the
acceptor SOM pool. N mineralization is defined as the difference between N obtained
from the donor pools and N stoichiometric demand of the acceptor pool (Parton et al.,
1988; Metherell et al., 1993). In this way, net N mineralization occurs when the donor
pool has low C:N ratio, but N is immobilized (taken up by microbes) when the donor
pool has high C:N ratio.

2.2 Optimal CUE

To quantify how microbial CUE varies along gradients of nutrient

availability, it can be hypothesized that microorganisms maximize their growth rate,
and hence their ecological competitiveness, by adapting resource (C and nutrients) use
efficiencies. This follows the growth maximization hypothesis (Mooshammer *et al.*,
2014; Manzoni *et al.*, 2017). Based on this hypothesis, Manzoni *et al.* (2017)
formulated a theoretical model expressing microbial CUE as a function of the
stoichiometric difference between decomposers and their substrate. The CUE for
which growth rate is maximized is the optimal CUE (*CUE_opt*) given by:
$$CUE_{opt} = CUE_{max} \times min\left\{1, \frac{CN_D}{CUE_{max}} \times \left[\frac{1}{CN_S} + \frac{I_N}{U_0}\right]\right\} \qquad (1)$$
where *CUE_max* is the maximum microbial CUE (dimensionless) when growth is
limited by C from the organic substrate. $CN_D$ and $CN_S$ are the C:N ratio (in mass,
dimensionless) of decomposer and their substrate, respectively. Although Manzoni *et*
*al.* (2017) indicated that mineral phosphorus (P) could also affect optimal CUE we
only considered N as a limiting nutrient. $I_N$ (g N kg$^{-1}$ soil) is the maximum rate at
which mineral N can be taken up by microbes, and $U_0$ (g C kg$^{-1}$ soil) is the C-limited
uptake rate (corresponding to the decomposition rate at optimal mineral N
concentration). When litter C:N is low or soil mineral N is in excess, the second term





in the minimum function (Eq. (1)) is higher than one, and $CUE_{opt} = CUE_{max}$ (C limited
conditions, as in nutrient-rich litter). In contrast, when mineral N is scarce, $CUE_{opt}$
decreases with increasing substrate C:N ratio (N limited conditions, N-poor litter).
Lack of N in the organic substrates can be compensated by mineral N being
immobilized by microorganisms from the soil solution. Immobilization meets the
nutrient demands as long as it is lower than the maximum supply rate $I_E$, at which
point microbial CUE starts being down regulated. Thus, for any given C:N ratio in the
substrate, $CUE_{opt}$ increases with inorganic N concentration in the soil solution until
$CUE_{max}$ is reached. It should also be noted that Eq. (1) is interpreted at the microbial
community scale, not for individual organism.

2.3 Adaption of the optimal CUE model in the CENTURY model

CUE of decomposition ($CUE_d$) is also assumed to be equivalent to microbial

CUE in this study. Then we followed the theory from Manzoni *et al.* (2017) (Eq. (1))
to parameterize $CUE_d$ during litter decomposition into CENTURY (Fig. 1). Due to the
implicit representation of microbial growth in CENTURY, we replaced the original
optimality CUE model (Eq. (1)) by a simpler equation that involves the C:N ratios of
the donor and acceptor pools, rather than microbial C:N ratios:
$$CUE_{opt} = CUE_{max} \times \min\left(1, \left(\frac{CN_{lit}}{CN_{SOM}}\right)^a\right) \qquad (2)$$
Where $CN_{lit}$ and $CN_{SOM}$ are the C:N ratio (dimensionless) of litter (metabolic or
structural) and SOM pools (active, slow or passive), respectively. The The C:N ratio
of SOM (around 9:1 on a mass basis in CENTURY) is representative of the
decomposer biomass, its value being between the C:N ratios of the two major group
decomposers, soil microbes (7.4:1) (Cleveland and Liptzin, 2007) and soil fungi
(13.4:1, Zhang *and* Elser, 2017). $CUE_{max} = 0.8$ (dimensionless) is the maximum $CUE_d$
achieved when nutrients are not limiting (Manzoni *et al.*, 2012; Sinsabaugh *et al.*,
2013) and *a* (g N kg$^{-1}$ soil) is an exponent capturing the effect of mineral N uptake by
microbes on $CUE_d$. $CUE_d$ being expected to increase with mineral N availability (Eq.
(1)), *a* is assumed to be a linear function of the mineral N concentration ($N_{min}$, g N kg$^{-1}$





soil):
$$a = m_1 \times (N_{min} - n_1) \qquad\qquad (3)$$
$m_1$ (kg g$^{-1}$ N) and $n_1$ (g N kg$^{-1}$ soil) are two coefficients that need to be calibrated. Eqs.
(2) and (3) modulate the decrease in CUE$_d$ with decreasing litter quality when mineral
N availability changes – the exponent $a$ increases with increasing mineral N
availability, causing an increase in CUE$_d$ at any given litter C:N ratio. Hence,
increasing $a$ value mimics an increase in $I_N$ in Equation1. Fig. 2a illustrates how CUE$_d$
from Eq. (2) varies as a function of mineral N concentration, for different values of
litter C:N.

Eqs. (2) and (3) were implemented in CENTURY to modify the originally

fixed $CUE_d$ (Fig. 1). With this change, the fractions of C from litter that remain in
SOM are all mediated by stoichiometric constraints and mineral N availability, at the
expense of additional parameters to fit.

2.4 Constraint of soil nutrient availability on litter decomposition rate

CENTURY is a first-order decay model in which decomposition rates of

metabolic and structural litter are modulated by scaling factors of soil temperature
($f(tem)$) and moisture ($f(water)$) (Parton $et\ al.$, 1988). Here, we introduced an
additional mineral N scaling factor ($f(N_{min})$, 0–1, dimensionless) to account for the
limitation of very low mineral N availability on litter decay rate ( $D(C_{lit})$).
$$D(C_{lit}) = C_{lit} \times k \times f(tem) \times f(water) \times f(N_{min}) \qquad (4)$$
where $C_{lit}$ is the C (g C kg$^{-1}$ soil) in litter pool (metabolic or structural). $k$ is the
potential maximum turnover rate (day$^{-1}$) at optimal soil temperature, moisture and
nutrient conditions. We assumed that the scaling factor of mineral N increases linearly
with increasing soil mineral N concentration ($N_{min}$, Eq. (5)) below a threshold value of
$1/m_2$ g N kg$^{-1}$ soil, where $m_2$ is a positive coefficient which needs to be calibrated (Fig.
2b). The inhibition effect of mineral N only occurs in case of immobilization (1/$CN_{lit}$
< $CUE_{opt}/CN_{SOM}$). The specific function $f(N_{min})$ can be expressed as:



$$f(N_{min}) = \begin{cases} \min(1, m_2 \times N_{min}) \,, & \frac{cue_{opt}}{CN_{SOM}} - \frac{1}{CN_{lit}} > 0 \\ 1 & , & \frac{cue_{opt}}{CN_{SOM}} - \frac{1}{CN_{lit}} \le 0 \end{cases} \qquad (5)$$


2.5 Model parameterization and validation

To determine the respective impacts of including flexible $CUE_d$ and N

availability constraining decay rates, we built four conceptual litter decay models
(Table 1). Model M0 corresponds to the default CENTURY parameterization of a
fixed $CUE_d$ and no constraints of N availability on litter decay rates ($f(N_{min}) = 1$).
Model M1 accounts for flexibility in CUE from Eq. (2) and N constraints on decay
rates by Eq. (5). Model M2 has flexible $CUE_d$ but no N constraints on decay rates
($f(N_{min}) = 1$). Model M3 has N constraints on decay rates but a fixed $CUE_d$ (Table 1).
All of these four models are run at a daily time step. This range of models allows
identifying which mechanisms are at play during decomposition – flexible $CUE_d$ only
(M3), mineral N limitation only (M2), both mechanisms (M1), or none (M0).

For calibrating model parameters and evaluation of their results, we collected

data of laboratory litter incubation experiments from Recous *et al*. (1995) (5
experiments) and Guenet *et al*. (2010) (9 experiments, Table A2). The incubation
experiments of Recous *et al*. (1995) and Guenet *et al*. (2010) continued 80 and 124
days, respectively. Recous *et al*. (1995) used corn residues (C:N = 130) and Guenet *et*
*al*. (2010) used wheat straw (C:N = 44) in their incubation experiments. The C:N
ratios of those corn residue and wheat straw span the range of litter C:N ratios among
different ecosystems (Manzoni *et al*., 2012;
https://www.planetnatural.com/composting-101/making/c-n-ratio/). In the incubation
experiments, plant litter was firstly cut into fine fragments before it was mixed with
mineral soil. Soil temperature and moisture condition were kept constant during the
experiment. Respired C from the incubated litter and SOC, as well as the soil mineral
N concentrations were measured continuously across the incubation period. More
detailed information about the incubation experiments of Recous *et al*. (1995) and
Guenet *et al*. (2010) can be found in Table A2.

The initial C storage and C:N ratios of litter and SOM pool, as well as soil



temperature and moisture condition for decomposition in all of the four version
models (M0-M3) were set based on observations (Table A2). In M1 and M4 model,
the observed mineral N concentrations across the incubation period were used to
calculate daily N inhibition effect (Eq. (5)). The observed cumulative respired litter-C
(g C kg$^{-1}$ soil) measured in the incubation experiments was used to calibrate the model
parameter values. Parameter calibration was performed for each model with the
shuffled complex evolution (SCE) algorithm developed by Duan *et al.*, (1993). The
SCE algorithm relies on a synthesis of four concepts that have proved successful for
global optimization: combination of probabilistic and deterministic approaches;
clustering; systematic evolution of a complex of points spanning the space in the
direction of global improvement and competitive evolution (Duan *et al.*, 1993). More
detailed description of this SCE optimization method can be found in Duan *et al.*
(1993, 1994). In this study, the RMSE (root mean square error, Eq. (6)) between
simulated and measured cumulative respired litter-C (%) on all observation days
(Table A2) of each incubation experiment was used as the objective function, and the
parameters minimizing RMSE between simulated and observed cumulative respired
litter-C were regarded as optimal parameter values.
$$RMSE = \sqrt{\left(\frac{\sum_{i=1}^{n}(Sim_i - Obs_i)^2}{n}\right)} \qquad (6)$$
where $n$ is the number of observation days, $Sim_i$ and $Obs_i$ (%) are the simulated
and observed percent of cumulative litter-C flux on day $i$, respectively.
We used leave-one-out cross-validation (Kearns and Ron, 1997; Tramontana
*et al.*, 2016) to evaluate each of the four models (i.e. M0-M3), a cross validation
method used when data is scarce. The number of cross-validations corresponds to the
number of incubation experiments (14). Each time, one of the 14 incubation
experiments was left out as the validation sample, and the remaining 13 experiments
were used to train model parameters. In addition to RMSE, we also adopted the
Akaike Information Criterion (AIC, Bozdogan, 1987, Eq. (7)) to determine the
relative quality of the four version models on estimating cumulative respired litter-C.
$$AIC = n \times ln\left(\frac{\sum_{i=1}^{n}(Sim_i - Obs_i)^2}{n}\right) + 2n_p \qquad (7)$$



where $n_p$ is the number of model parameters. The evaluation of AIC is important here
because depending on the model M1, M2, or M3 parameters have to be determined
(Table 1), requiring us to weigh both model accuracy and robustness.

2.6 Impacts of litter stoichiometry and mineral N availability on SOM accumulation
We used the model M1, with flexible $CUE_d$ and decomposition rate function
of available N to study the impacts of litter stoichiometry (C:N ratio) and soil mineral
N availability on the formation and accumulation of SOM. Totally, 24 idealized
simulation experiments with different values of litter C:N ratios and soil mineral N
availabilities were conducted (Table A3). The assumed litter C:N ratios ($CN_{lit}$) of 10,
15, 30, 60, 120 and 200 span the variation among most natural substrates and soil
amendments from organic matter input in agriculture (Manzoni *et al.*, 2012;
https://www.planetnatural.com/composting-101/making/c-n-ratio/). The assumed
range of mineral N availability ($N_{min}$) of 0.001, 0.005, 0.01 and 0.05 g N kg$^{-1}$ soil span
the observed concentrations of soil mineral N in major terrestrial ecosystems
(Metherall *et al.*, 1993).
In each simulation experiment, M1 was run for 5000 years to bring the litter
and SOM pools in equilibrium with the prescribed litter input flux. The daily input rate
of plant litter was set to 0.006 g C kg$^{-1}$ soil day$^{-1}$, and the initial C stock of litter and
SOM pools were all set to be 0 g C kg$^{-1}$ soil. During the simulation, soil temperature
and soil water content were assumed to be 25 °C and 60% of water holding capacity,
respectively. We emphasized that our goal with this simplified scenario was to single
out the effects of stoichiometric constraints, not to simulate the effects of a realistic
climatic regime. Parameter values for M1 (with $m_1 = 0.54$, $n_1 = 0.50$ and $m_2 = 296.8$)
used here were optimized based on all of the 14 incubation experiments from Recous
*et al.* (1995) and Guenet *et al.* (2010) (see above). More detailed information about
the specific settings of our simulation experiments can be found in Table A3.

**3 Results**





3.1 Evaluation of different models

Results of leave-one-out cross-validation suggest that model M1 provides

more accurate prediction of cumulative respired litter-C than other models (Fig. 3).

The differences between simulated and observed cumulative respired litter-C from

M1 are mostly (over 93% of the data) less than 6% (Fig. S1b in supplementary

materials). The average RMSE of predicted cumulative respired litter-C from M1

(3.0%) is lower than that of model M0 (4.1%). Models M2 and M3 have slightly

lower RMSE values than M0 (3.7% and 3.8%, respectively) but perform worse than

M1 (Fig. 4). However, the average AIC of all the models are comparable, suggesting

that models with more fitted parameters do not over-fit the observations (Fig. 4).

Model M1 captures the differences in respiration rates due to different C:N

ratios of substrate and varying levels of mineral N availability across the 14

incubation experiments (Fig. 5). While model M3 can reproduce the observed effect

of soil mineral N availability on litter respirations rates (Fig. 5d), it underestimate the

cumulative respired $CO_2$ from low quality litter ($CN_{lit} = 130$) at high mineral N

concentrations (> 0.04 g N kg$^{-1}$ soil). Models M0 and M2 cannot represent the effects

of soil mineral N on litter respiration rate (Figs. 5a, c), and their predictions are more

biased from the observed values compared to M1. In addition, model M1 can also

capture the temporal evolution of cumulative respired litter-C in different incubation

experiments (Fig. 5b).

The predicted CUE$_d$ of decomposed litter and the limitation effects of soil

mineral N availability on litter decay rate from the $f(N_{min})$ function (Eq. (5)) are

different among the four tested models (Fig. A2). In models M0 and M3, which used a

fixed CUE$_d$, the fitted values of CUE$_d$ calculated with optimized parameters during

the incubation period are about 0.57 and 0.54, respectively (Figs. A2a, d). In models

M1 and M2, the CUE$_d$ varies with the C:N ratios of plant litter, and is only slightly

affected by soil mineral N concentrations (Figs. A2b, c). For very low quality litter

with a C:N ratio of 130, the CUE$_d$ in models M1 and M2 are 0.55 and 0.56,

respectively, which are higher than for better quality litter with C:N ratio of 44

(approximately 0.40 and 0.44 in M1 and M2, respectively). CUE$_d$ from Eq. (2)





calibrated with the data of the two incubation experiments, decreases with increasing
$CN_{lit}/CN_{SOM}$ (Fig. 6). The average $CUE_d$ value is larger than the average of data
compiled for microbial CUE of litter decomposition in terrestrial ecosystems by
Manzoni *et al*. (2017). This is shown by the gray circles in Fig. 6. Our optimized
values of $CUE_d$ for a given C:N ratio are more comparable with microbial CUE
observed in incubations of soil mixed with litter (Gilmour and Gilmour, 1985;
Devêvre and Horwáth, 2000; Thiet *et al*., 2006), shown as black squares in Fig. 6.
Models M0 and M2 do not include the N inhibition effects on litter decay rate, thus
the $f(N_{min})$ in these two models is always 1 (Figs. A2e, g). In M1 and M3, the N
inhibition effect changes with both the litter C:N ratio and the mineral N availability
(Figs. A2f, h).

3.2 The effect of litter quality *vs* quantity on equilibrium SOM stocks

Model M1 predicts that the size of the SOM pool at equilibrium is mainly

determined by litter stoichiometry, with a minor effect of soil mineral N (Fig. 7). The
lower C:N ratio of litter is, the higher equilibrium SOC stock. For litter with a specific
C:N ratio, high soil mineral N concentration (e.g. above 0.05 g N kg$^{-1}$ soil) generally
produces a slightly larger equilibrium SOC stock than a low mineral N concentration
(Fig. 7). Further analysis suggests that the SOC at equilibrium increases with
decreasing litter C:N because the SOC pool is positively related to the $CUE_d$; however
the limitation of soil mineral N on litter decomposition rate almost shows no impact
on SOC (Fig. A3).

**4 Discussion**

We hypothesized that stoichiometric constraints (flexible $CUE_d$ or inhibition

of decomposition under N limited conditions) played a role in shaping the trajectory
of litter decomposition, with potential consequences on predicted SOC stocks. Our
results suggest that with flexible $CUE_d$ and the inhibition effects of soil mineral N on
litter decay rate, the model M1 developed from CENTURY can be a reliable tool for





predicting litter decomposition. Evaluation of the model (M1) using data from
incubation experiments indicate that this modified model captures the effect of
variable litter quality (stoichiometry) and mineral N availability on respiration rates
(Fig. 5), without strongly inflating the complexity of CENTURY (Table 1). As the
stoichiometric constraints are implemented in the generalizable and widely used
structure of CENTURY and require only three parameters to be calibrated, they can
also be easily implemented into land surface models for large spatial scale
applications.

Accurately representing N control of microbial processes during litter

decomposition has been suggested to be important for modeling the connection
between the litter inputs, $CUE_d$, and soil C dynamics (Gerber *et al.*, 2010; Manzoni *et
al.*, 2012; Cotrufo *et al.*, 2013; Sinsabaugh *et al.*, 2013). In model M1, soil mineral N
affects the litter-C flux via two mutually different pathways: (1) mineral N availability
affects the litter decay rate and (2) flexible $CUE_d$ determining the partition of
decomposed C into SOC products and respired $CO_2$ (Fig. 1). Therefore, an increase in
soil mineral N concentration enhances litter decay rates, which alone will increase the
flux of litter-derived $CO_2$ (Eq. (5) and Fig. A4). However, as higher N concentration
also results in a higher $CUE_d$ (Eq. (2)), more C is transferred to SOC and less C is
respired. In this way, SOC is predicted to accumulate with increasing mineral N
availability when using model M1 (Fig. 7).

Moreover, describing N limitations on both the decomposition rate and

flexible $CUE_d$ might allow our model to explain the observed diverse responses of
litter respiration rate to added mineral N in fertilization experiments (Hobbie and
Vitousek, 2000; Guenet *et al.*, 2010; Janssens *et al.*, 2010). In these experiments, the
net changes in respiration rate depend on the combined effects of added N on litter
decay rate and $CUE_d$ of the decayed litter (Fig. A4).

Existing studies have adopted approaches that differ from our definition to

explicitly represent the N inhibition effects on microbial processes (Eq. (5)) (Manzoni
and Porporato, 2009; Bonan *et al.*, 2013; Fujita *et al.*, 2014; Averill and Waring, 2018).
In these previous studies, $f(N_{min})$ was assumed equal to the ratio between immobilized



mineral N and the N deficit for keeping the stoichiometric balance (i.e. C:N) of
decomposer biomass or other receiver pools. Using the notation of Section 2, this
constant can be expressed as:

$$
f(N_{min}) = \begin{cases} \min\left(1, \dfrac{m_3 \times N_{min}}{U_0 \times \left(\dfrac{CUE_{opt}}{CN_{SOM}} - \dfrac{1}{CN_{lit}}\right)}\right), & \dfrac{CUE_{opt}}{CN_{SOM}} - \dfrac{1}{CN_{lit}} > 0 \\ 1, & \dfrac{CUE_{opt}}{CN_{SOM}} - \dfrac{1}{CN_{lit}} \le 0 \end{cases} \tag{8}
$$

where $m_3$ is a coefficient that needs to be optimized. $U_0$ (g C kg$^{-1}$ soil day$^{-1}$) is the C
uptake rate (equivalent to the litter decomposition rate in absence of leaching) when
soil mineral N is fully adequate for litter decay (i.e. $f(N_{min}) = 1$), and can be calculated
as:

$$
U_0 = C_{lit} \times k \times f(tem) \times f(water) \tag{9}
$$

We have tested this formulation in the CENTURY-based model, in addition to the
other formulations (Table 1). The model with Eq. (8) gave a more biased estimation
on cumulative respired litter-C than the model using Eq. (5) (Fig. A5). We surmise
that although Eq. (8) can better represent the underlying microbial mechanisms of N
inhibition effects, it also increases the model complexity and in turn the efforts and
uncertainty in model parameterization.

The importance of litter quality for SOM formation as found here is in line

with recent experiments (Bahri *et al.*, 2008; Rubino *et al.*, 2010; Walela *et al.*, 2014)
and modeling studies (Grandy and Neff, 2008; Cotrufo *et al.*, 2013). SOM is mainly
formed though the partial decomposition of plant debris by microorganisms (Paul,
2007; Knicker, 2011; Cotrufo *et al.*, 2013). The conceptual model developed by
Cotrufo *et al.* (2013) suggested that although labile litter was decomposed faster than
recalcitrant litter, a higher fraction of this labile litter-C would be incorporated into
microbial biomass and subsequently incorporated into SOM pool (corresponding to a
higher CUE$_d$). Therefore, labile litter inputs tend to form a larger SOM pool than the
poor-quality (high C:N ratio) litter that is generally used by microbes at lower
efficiency. Our simulations of decomposition process of plant litter with different C:N
ratios also suggest that litter of good quality (with low C:N ratio) can induce a larger

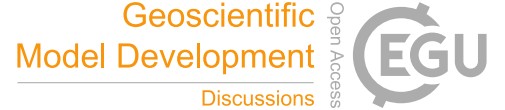



SOM pool than the poor-quality litter (Fig. 7). $CUE_d$ plays a more important role than
the inhibition effect of low mineral N concentration in determining the size of the
stable SOM pool (Fig. A3).

The predictions from Cotrufo *et al*. (2013) and this study contrasts with the

conventional hypothesis whereby the poor-quality litter with low decay rate and small
$CUE_d$ are preferential to be accumulated in SOM (Berg and Mcclaugherty, 2008;
Walela *et al.*, 2014). This view of SOM stabilization, however, seems to apply to
N-limited systems with high C:N litter and where microbial remains are recalcitrant to
decomposition (e.g., boreal forests) – in these systems SOC does accumulate despite
its low quality (Kyaschenko et al. 2017). Moreover, one could argue that higher $CUE_d$
implies larger microbial biomass, allowing faster decomposition (Allison et al., 2010).
These feedbacks between microbial biomass and decomposition rate were not
implemented in the current model, but could offer additional flexibility – again at the
expense of more difficult model parameterization.

The $CUE_d$ formulation from Eq. (2) with parameters calibrated from the two

sets of incubation experiments might underestimate the impacts of litter quality on
microbial CUE under natural conditions, in particular in case of SOM decomposition.
In both incubation experiments, litter is firstly cut into fine fragments and then fully
mixed with mineral soil (Recous *et al*.,1995; Guenet *et al*., 2010). Thus, the nutrient
accessibility, air permeability and some other environmental factors (e.g. pH) of
incubated litter are different from those of decaying litter in more natural,
heterogeneous soil conditions. Those different decomposition conditions might be
responsible for the differences observed in Fig. 6 between our CUE estimates and
previously reported values. We speculate that more heterogeneous conditions reduce
nutrient availability and thus might cause lower CUE. Similarly, CUE of surface litter
decomposers may be lower than we estimated because litter not mixed with soil is
probably subject to strongly nutrient limitation.

This study provides some insights on processes leading to increased SOM

sequestration. Soil C sequestration plays a crucial role in food security and land $CO_2$
emission (Lal, 2004). The international initiative '4 per 1000' has been proposed to

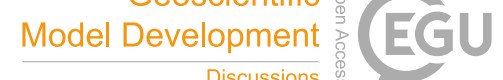

increase global SOM stock by 0.4% per year to compensate for anthropogenic $CO_2$
emissions (Baveye *et al.*, 2018). Transforming more plant litter into stable SOM (e.g.
humic substances) has been suggested as an effective strategy to sequester more C in
soil (Prescott, 2010). Our model results show a positive linear relationship between
equilibrium SOC stock and CUE of decomposed litter (Fig. A3). This result can also
be interpreted by calculating the analytical equilibrium SOC storage of a fully linear
model including only one litter pool and one SOC pool. In such a model, SOC
receives C from the litter at a rate $CUE_d \times D$, where $D$ is the litter decomposition rate,
which equals to litterfall at steady state. SOC is lost via first order decay with a decay
constant $k$. At steady state, input to and outputs from the SOC pool are equal and thus,
$$\mathrm{CUE_d} \times D = k \times \mathrm{SOC} \;\rightarrow\; \mathrm{SOC} = \mathrm{CUE_d}\frac{D}{k} \tag{10}$$
With a mean residence time of C in the SOC between 10 and 20 years and $D$
approximated by litterfall (Table A3), SOC at equilibrium is predicted to scale linearly
with $CUE_d$, with a slope approximately between 20 and 40, consistent with results in
Fig. A3.
Therefore, litter quality needs to be controlled to maximize C sequestration
in SOM pool (Eq. (2)). In line with previous studies (Prescott, 2010; Smith, 2016),
our model predicts that adding N through fertilization and N-fixing plants will not
only increase litter decay but also the fraction of litter-C being transformed into SOM
and ultimately SOC stocks. However, application of mineral N fertilizer is associated
with risk not considered here, like increasing land $N_2O$ emission (Mosier and Kroeze,
2000; Kanter *et al*., 2016; Yi *et al*., 2017) and causing nitrate leaching which in turn
can induce water pollution (Cao *et al*., 2006; Strokal *et al*., 2016). Due to the negative
environmental impacts of mineral N addition, the use of N-rich litter substrates for
increasing SOM is advised.
Further validation and development of our model are still necessary to
decrease the model uncertainties. Soil mineral N which affects both litter decay rate
and CUE of decayed litter is seldom monitored in litter incubation experiments (e.g.
Walela *et al*., 2014; Stewart *et al*., 2015) and field litter decay experiments (e.g. Gholz



*et al*., 2000; Harmon *et al*., 2009), with few exceptions (Recous *et al*., 1995; Guenet *et*
*al*., 2010). An increasing number of land surface models (e.g. ORCHIDEE-CNP, Goll
*et al*., 2017) have representations of the terrestrial N cycle. By incorporating our litter
decomposition formulation in these land surface models that simulate the dynamics of
soil mineral N concentration, it will be possible to test and validate our developments
with more extensive data from laboratory and field experiments. Moreover, similar to
N, P has also been suggested as another important factor for litter decomposition and
SOM formation (Güsewell and Verhoeven, 2006; Talkner et al., 2009; Manzoni et al.,
2010; Prescott, 2010), especially in regions with highly weathered soil (Goll *et al*.,
2012, 2017; Yang *et al*., 2014). So it might be necessary to include the effects of P on
litter decay rate and $CUE_d$ into our model for further decrease the simulation
uncertainties.

## 5 Conclusions

By adapting the hypothesis of optimal microbial CUE proposed by Manzoni

et al. (2017) for use in a CENTURY-based model and also introducing a N scaling
function to represent the limits of mineral N availability on litter decay rate, we
developed a simple but effective litter decomposition model that accounts for key
stoichiometric constraints during decomposition. Validation using observation data
obtained from laboratory incubation experiments indicated that our model could well
predict the respiration rates of litter in different qualities at various levels of mineral N
availability. Idealized simulations using our model revealed that the quality of litter
inputs plays an important role in determining the soil C stock at equilibrium SOM
pool. High-quality litter (i.e. with low C:N ratio) tends to form a larger SOM pool as
it can be more efficiently utilized by microorganisms than recalcitrant litter (e.g. high
C:N ratio). Overall, the developed model captures the microbial mechanisms
mediating litter stoichiometry and soil mineral N effects on litter decomposition and
SOM formation – representing an improvement over most existing large-scale litter
decay models. Due to the simple and generalizable structure of our model, it can be





incorporated into existing land surface models for further long-term and large spatial
scale applications.



**Code and data availability**

The CENTURY-based model used here is programmed in MATLAB language. The source code is available online (https://github.com/hchzhang/CENYUTY_CUE/tree/v1.0, DOI: 10.5281/zenodo.1307384). All the data used in this study can be obtained from published literatures. Specific references of these data can be found in section 2.5.

**Competing interests**

The authors declare that they have no conflict of interest.

**Acknowledgements**

HZ, DSG, PC and YH are funded by the IMBALANCE-P project of the European Research Council (ERC-2013-SyG- 610028). SM acknowledges the support of the Swedish Research Council Vetenskapsrådet (grants 2016-04146 and 2016-06313).




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

©c Author(s) 2018. CC BY 4.0 License.





Fujita, Y., Witte, J.-P. M., and van Bodegom, P. M.: Incorporating microbial ecology concepts into

global soil mineralization models to improve predictions of carbon and nitrogen fluxes,

Global Biogeochemical Cycles, 28, 223-238, 10.1002/2013gb004595, 2014.

Garc ía-Palacios, P., McKie, B. G., Handa, I. T., Frainer, A., H ättenschwiler, S., and Jones, H.: The

importance of litter traits and decomposers for litter decomposition: a comparison of aquatic

and terrestrial ecosystems within and across biomes, Functional Ecology, 30, 819-829,

10.1111/1365-2435.12589, 2016.

Gerber, S., Hedin, L. O., Oppenheimer, M., Pacala, S. W., and Shevliakova, E.: Nitrogen cycling

and feedbacks in a global dynamic land model, GB1001, GLOBAL BIOGEOCHEMICAL

CYCLES, 2010.

Gholz, H. L., Wedin, D. A., Smitherman, S. M., Harmon, M. E., and Parton, W. J.: Long-term

dynamics of pine and hardwood litter in contrasting environments: toward a global model of

decomposition, Global Change Biology, 6, 751-765, 2000.

Gilmour, C. M., and Gilmour, J. T.: Assimilation of carbon by the soil biomass, Plant & Soil, 86,

101-112, 1985.

Goll, D. S., Brovkin, V., Parida, B. R., and Reick, C. H.: Nutrient limitation reduces land carbon

uptake in simulations with a model of combined carbon, nitrogen and phosphorus cycling,

Biogeosciences Discussions, 9, 3547-3569, 2012.

Goll, D. S., Vuichard, N., Maignan, F., Jornet-Puig, A., Sardans, J., Violette, A., Peng, S., Sun, Y.,

Kvakic, M., and Guimberteau, M.: A representation of the phosphorus cycle for ORCHIDEE

(revision 4520), Geoscientific Model Development, 10, 3745-3770, 2017.

Grandy, A. S., and Neff, J. C.: Molecular C dynamics downstream: the biochemical decomposition

sequence and its impact on soil organic matter structure and function, Science of the Total

Environment, 404, 297-307, 2008.

Guenet, B., Neill, C., Bardoux, G., and Abbadie, L.: Is there a linear relationship between priming

effect intensity and the amount of organic matter input?, Applied Soil Ecology, 46, 436-442,

10.1016/j.apsoil.2010.09.006, 2010.

G üsewell, S., and Verhoeven, J. T. A.: Litter N:P ratios indicate whether N or P limits the

decomposability of graminoid leaf litter, Plant & Soil, 287, 131-143, 2006a.

G üsewell, S., and Verhoeven, J. T. A.: Litter N:P ratios indicate whether N or P limits the



decomposability of graminoid leaf litter, Plant and Soil, 287, 131-143,
10.1007/s11104-006-9050-2, 2006b.
Hansen, S., Jensen, H. E., Nielsen, N. E., and Svendsen, H.: Simulation of nitrogen dynamics and
biomass production in winter wheat using the Danish simulation model DAISY, Fertilizer
Research, 27, 245-259, 1991.
Harmon, M. E., Silver, W. L., Fasth, B., Chen, H. U. A., Burke, I. C., Parton, W. J., Hart, S. C.,
and Currie, W. S.: Long-term patterns of mass loss during the decomposition of leaf and fine
root litter: an intersite comparison, Global Change Biology, 15, 1320-1338,
10.1111/j.1365-2486.2008.01837.x, 2009.
Hobbie, S. E., and Vitousek, P. M.: Nutrient limitation of decomposition in Hawaiian forests,
Ecology, 81, 1867-1877, 2000.
Huang, Y., Guenet, B., Ciais, P., Janssens, I. A., Soong, J. L., Wang, Y., Goll, D., Blagodatskaya,
E., and Huang, Y.: ORCHIMIC (v1.0), a microbe-driven model for soil organic matter
decomposition designed for large-scale applications, Geoscientific Model Development
Discussions, 1-48, 10.5194/gmd-2017-325, 2018.
Ingwersen, J., Poll, C., Streck, T., and Kandeler, E.: Micro-scale modelling of carbon turnover
driven by microbial succession at a biogeochemical interface, Soil Biology and Biochemistry,
40, 864-878, 10.1016/j.soilbio.2007.10.018, 2008.
Janssens, I. A., Dieleman, W., Luyssaert, S., Subke, J. A., Reichstein, M., Ceulemans, R., Ciais, P.,
Dolman, A. J., Grace, J., Matteucci, G., Papale, D., Piao, S. L., Schulze, E. D., Tang, J., and
Law, B. E.: Reduction of forest soil respiration in response to nitrogen deposition, Nature
Geoscience, 3, 315-322, 10.1038/ngeo844, 2010.
Kätterer, T., and Andrén, O.: The ICBM family of analytically solved models of soil carbon,
nitrogen and microbial biomass dynamics — descriptions and application examples,
Ecological Modelling, 136, 191-207, 2001.
Kanter, D. R., Zhang, X., Mauzerall, D. L., Malyshev, S., and Shevliakova, E.: The importance of
climate change and nitrogen use efficiency for future nitrous oxide emissions from
agriculture, Environmental Research Letters, 11, 094003, 10.1088/1748-9326/11/9/094003,

2016.

Kearns, M. & Ron, D. (1997) Algorithmic stability and sanity-check bounds for leave-one-out





cross-validation. Neural Computation, 11, 1427-1453.
Knicker, H.: Soil organic N - An under-rated player for C sequestration in soils?, Soil Biology &
Biochemistry, 43, 1118-1129, 2011.
Krinner, G., Viovy, N., de Noblet-Ducoudré, N., Ogée, J., Polcher, J., Friedlingstein, P., Ciais, P.,
Sitch, S., and Prentice, I. C.: A dynamic global vegetation model for studies of the coupled
atmosphere-biosphere system, Global Biogeochemical Cycles, 19,
doi:10.1029/2003GB002199, 10.1029/2003gb002199, 2005.
Kucharik, C. J., Foley, J. A., Delire, C., Fisher, V. A., Coe, M. T., Lenters, J. D., Young-Molling,
C., Ramankutty, N., Norman, J. M., and Gower, S. T.: Testing the performance of a dynamic
global ecosystem model: Water balance, carbon balance, and vegetation structure, Global
Biogeochemical Cycles, 14, 795-825, 10.1029/1999gb001138, 2000.
Kyaschenko, J., Clemmensen, K. E., Karltun, E., and Lindahl, B. D.: Below-ground organic
matter accumulation along a boreal forest fertility gradient relates to guild interaction within
fungal communities, Ecology letters, 20, 1546-1555, 10.1111/ele.12862, 2017.
Lal, R.: Soil carbon sequestration impacts on global climate change and food security, Science,
304, 1623-1627, 10.1126/science.1097396, 2004.
Lekkerkerk, L., Lundkvist, H., Ågren, G. I., Ekbohm, G., and Bosatta, E.: Decomposition of
heterogeneous substrates; An experimental investigation of a hypothesis on substrate and
microbial properties, Soil Biology & Biochemistry, 22, 161-167, 1990.
Liski, J., Palosuo, T., Peltoniemi, M., and Sievänen, R.: Carbon and decomposition model Yasso
for forest soils, Ecological Modelling, 189, 168-182, 10.1016/j.ecolmodel.2005.03.005, 2005.
Luo, Y., Ahlström, A., Allison, S. D., Batjes, N. H., Brovkin, V., Carvalhais, N., Chappell, A.,
Ciais, P., Davidson, E. A., and Finzi, A.: Toward more realistic projections of soil carbon
dynamics by Earth system models, Global Biogeochemical Cycles, 30, n/a-n/a, 2016.
Manzoni, S., Jackson, R. B., Trofymow, J. A., and Porporato, A.: The global stoichiometry of litter
nitrogen mineralization, Science, 321, 684-686, 2008.
Manzoni, S., and Porporato, A.: Soil carbon and nitrogen mineralization: Theory and models
across scales, Soil Biology and Biochemistry, 41, 1355-1379, 10.1016/j.soilbio.2009.02.031,

2009.

Manzoni, S., Trofymow, J. A., Jackson, R. B., and Porporato, A.: Stoichiometric controls on



carbon, nitrogen, and phosphorus dynamics in decomposing litter, Ecological Monographs,

80, 89-106, 2010.

Manzoni, S., Taylor, P., Richter, A., Porporato, A., and Agren, G. I.: Environmental and
stoichiometric controls on microbial carbon-use efficiency in soils, The New phytologist, 196,
79-91, 10.1111/j.1469-8137.2012.04225.x, 2012.
Manzoni, S., Capek, P., Mooshammer, M., Lindahl, B. D., Richter, A., and Santruckova, H.:
Optimal metabolic regulation along resource stoichiometry gradients, Ecology letters, 20,
1182-1191, 10.1111/ele.12815, 2017.
Metherall, A. K., Harding, L. A., Cole, C. V., and Parton, W. J.: CENTURY Soil Organic Matter
Model Environment Technical Documentation, Agroecosystem Version 4.0, Great Plains
System Research Unit, Technical Report No. 4. USDA-ARS, Ft. Collins., 1993.
Molina, J. A. E., Clapp, C. E., Shaffer, M. J., Chichester, F. W., and Larson, W. E.: NCSOIL, A
Model of Nitrogen and Carbon Transformations in Soil: Description, Calibration, and
Behavior1, Soil Science Society of America Journal, 47, 85-91, 1983.
Moorhead, D. L., and Sinsabaugh, R. L.: A Theoretical Model of Litter Decay and Microbial
Interaction, Ecological Monographs, 76, 151-174, 2006.
Mooshammer, M., Wanek, W., Hammerle, I., Fuchslueger, L., Hofhansl, F., Knoltsch, A.,
Schnecker, J., Takriti, M., Watzka, M., Wild, B., Keiblinger, K. M., Zechmeister-Boltenstern,
S., and Richter, A.: Adjustment of microbial nitrogen use efficiency to carbon:nitrogen
imbalances regulates soil nitrogen cycling, Nature communications, 5, 3694,
10.1038/ncomms4694, 2014.
Mosier, A., and Kroeze, C.: Potential impact on the global atmospheric N2O budget of the
increased nitrogen input required to meet future global food demands, Chemosphere - Global
Change Science, 2, 465-473, 2000.
Pagel, H., Ingwersen, J., Poll, C., Kandeler, E., and Streck, T.: Micro-scale modeling of pesticide
degradation coupled to carbon turnover in the detritusphere: model description and sensitivity
analysis, Biogeochemistry, 117, 185-204, 10.1007/s10533-013-9851-3, 2013.
Parton, W., Silver, W. L., Burke, I. C., Grassens, L., Harmon, M. E., Currie, W. S., King, J. Y.,
Adair, E. C., Brandt, L. A., Hart, S. C., and Fasth, B.: Global-scale similarities in nitrogen
release patterns during long-term decomposition, Science, 315, 361-364,





738 10.1126/science.1134853, 2007.

739 Parton, W. J., Stewart, J. W. B., and Cole, C. V.: Dynamics of C, N, P and S in grassland soils: a

740  model, Biogeochemistry, 5, 109-131, 1988.

741 Paul, E. A.: Soil Microbiology, Ecology and Biogeochemistry, Academic Press, San Diego, CA,

742  USA., 2007.

743 Prescott, C. E.: Litter decomposition: what controls it and how can we alter it to sequester more

744  carbon in forest soils?, Biogeochemistry, 101, 133-149, 10.1007/s10533-010-9439-0, 2010.

745 Recous, S., Robin, D., Darwis, D., and Mary, B.: Soil inorganic N availability: Effect on maize

746  residue decomposition, Soil Biology & Biochemistry, 27, 1529-1538, 1995.

747 Rubino, M., Dungait, J. A. J., Evershed, R. P., Bertolini, T., Angelis, P. D., D'Onofrio, A.,

748  Lagomarsino, A., Lubritto, C., Merola, A., and Terrasi, F.: Carbon input belowground is the

749  major C flux contributing to leaf litter mass loss: Evidences from a 13 C labelled-leaf litter

750  experiment, Soil Biology & Biochemistry, 42, 1009-1016, 2010.

751 Schimel, J. P., and Weintraub, M. N.: The implications of exoenzyme activity on microbial carbon

752  and nitrogen limitation in soil: a theoretical model, Soil Biology & Biochemistry, 35,

753  549-563, 2003.

754 Schmidt, M. W., Torn, M. S., Abiven, S., Dittmar, T., Guggenberger, G., Janssens, I. A., Kleber, M.,

755  Kogel-Knabner, I., Lehmann, J., Manning, D. A., Nannipieri, P., Rasse, D. P., Weiner, S., and

756  Trumbore, S. E.: Persistence of soil organic matter as an ecosystem property, Nature, 478,

757  49-56, 10.1038/nature10386, 2011.

758 Sinsabaugh, R. L., Manzoni, S., Moorhead, D. L., and Richter, A.: Carbon use efficiency of

759  microbial communities: stoichiometry, methodology and modelling, Ecology letters, 16,

760  930-939, 10.1111/ele.12113, 2013.

761 Sitch, S., Smith, B., Prentice, I. C., Arneth, A., Bondeau, A., Cramer, W., Kaplan, J. O., Levis, S.,

762  Lucht, W., Sykes, M. T., Thonicke, K., and Venevsky, S.: Evaluation of ecosystem dynamics,

763  plant geography and terrestrial carbon cycling in the LPJ dynamic global vegetation model,

764  Global Change Biology, 9, 161-185, 10.1046/j.1365-2486.2003.00569.x, 2003.

765 Six, J., Frey, S. D., Thiet, R. K., and Batten, K. M.: Bacterial and Fungal Contributions to Carbon

766  Sequestration in Agroecosystems, Soil Science Society of America Journal, 70, 555--569,

767  2006.





Smith, P.: Soil carbon sequestration and biochar as negative emission technologies, Global Change
Biology, 22, 1315-1324, 2016.
Stewart, C. E., Moturi, P., Follett, R. F., and Halvorson, A. D.: Lignin biochemistry and soil N
determine crop residue decomposition and soil priming, Biogeochemistry, 124, 335-351,
10.1007/s10533-015-0101-8, 2015.
Strokal, M., Ma, L., Bai, Z., Luan, S., Kroeze, C., Oenema, O., Velthof, G., and Zhang, F.:
Alarming nutrient pollution of Chinese rivers as a result of agricultural transitions,
Environmental Research Letters, 11, 024014, 10.1088/1748-9326/11/2/024014, 2016.
Talkner, U., Jansen, M., and Beese, F. O.: Soil phosphorus status and turnover in central-European
beech forest ecosystems with differing tree species diversity, European Journal of Soil
Science, 60, 338-346, 2010.
Thiet, R. K., Frey, S. D., and Six, J.: Do growth yield efficiencies differ between soil microbial
communities differing in fungal:bacterial ratios? Reality check and methodological issues,
Soil Biology and Biochemistry, 38, 837-844, 10.1016/j.soilbio.2005.07.010, 2006.
Tramontana, G., Jung, M., Schwalm, C. R., Ichii, K., Campsvalls, G., Ráduly, B., Reichstein, M.,
Altaf Arain, M., Cescatti, A., and Kiely, G.: Predicting carbon dioxide and energy fluxes
across global FLUXNET sites with regression algorithms, Biogeosciences Discussions, 13,
1-33, 2016.

Verberne, E. L. J., Hassink, J., Willigen, P. D., Groot, J. J. R., and Veen, J. A. V.: Modelling
organic matter dynamics in different soils, Netherlands Journal of Agricultural Science Issued
by the Royal Netherlands Society for Agricultural Science, 38, 221-238, 1990.
Walela, C., Daniel, H., Wilson, B., Lockwood, P., Cowie, A., and Harden, S.: The initial
lignin:nitrogen ratio of litter from above and below ground sources strongly and negatively
influenced decay rates of slowly decomposing litter carbon pools, Soil Biology and
Biochemistry, 77, 268-275, 10.1016/j.soilbio.2014.06.013, 2014.
Wieder, W. R., Bonan, G. B., and Allison, S. D.: Global soil carbon projections are improved by
modelling microbial processes, Nature Climate Change, 3, 909-912, 10.1038/nclimate1951,
2013.

Wieder, W. R., Cleveland, C. C., Smith, W. K., and Todd-Brown, K.: Future productivity and
carbon storage limited by terrestrial nutrient availability, Nature Geoscience, 8, 441-444,





2015.

Yang, X., Post, W. M., Thornton, P. E., and Ricciuto, D. M.: The role of phosphorus dynamics in

tropical forests - a modeling study using CLM-CNP, Biogeosciences, 11, 14439-14473, 2014.

Yi, Q., Tang, S., Fan, X., Zhang, M., Pang, Y., Huang, X., and Huang, Q.: Effects of nitrogen

application rate, nitrogen synergist and biochar on nitrous oxide emissions from vegetable

field in south China, PloS one, 12, e0175325, 10.1371/journal.pone.0175325, 2017.

Zhang, C. F., Meng, F. R., Bhatti, J. S., Trofymow, J. A., and Arp, P. A.: Modeling forest leaf-litter

decomposition and N mineralization in litterbags, placed across Canada: A 5-model

comparison, Ecological Modelling, 219, 342-360, 10.1016/j.ecolmodel.2008.07.014, 2008.

Zhang, J., and Elser, J. J.: Carbon:Nitrogen:Phosphorus Stoichiometry in Fungi: A Meta-Analysis,

Frontiers in microbiology, 8, 1281, 2017.






**Table 1** The four version of the litter decomposition model used in this study. $cue_{fit}$ is
optimized value of CUE. $m_1$ and $n_1$ are the coefficients in Eq. (3), and $m_2$ is the
coefficients in Eq. (5).

| Model version | CUE | $f(N_{min})$ | Parameters |
|---|---|---|---|
| M0 | fixed | 1 | $cue_{fit}$ |
| M1 | Eqs. (2), (3) | Eq. (5) | $m_1, n_1, m_2$ |
| M2 | Eqs. (2), (3) | 1 | $m_1, n_1$ |
| M3 | fixed | Eq. (5) | $cue_{fit}, m_2$ |


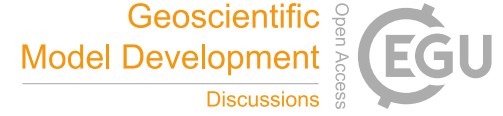



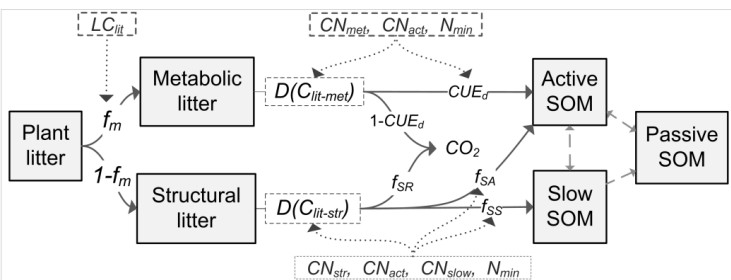


**Figure 1** Schematic diagram of the C flows in the litter decay model used in this study.

$f_m$ is the fraction of metabolic compounds in plant litter. $D(C_{lit\text{-}met})$ and $D(C_{lit\text{-}str})$ are

the decomposition rates (g C kg$^{-1}$ day$^{-1}$) of metabolic or structural litter, respectively.

$LC_{lit}$ is the lignin:C ratio (on a mass basis) of plant litter; $CN_{met}$, $CN_{str}$, $CN_{act}$, and

$CN_{slow}$ are the C:N ratio of metabolic litter pool, structural litter pool, active SOM

pool and slow SOM pool, respectively; $N_{min}$ is the concentration of mineral N in

solution (g N kg$^{-1}$ soil); $CUE_d$ is C use efficiency of the transformation from litter to

soil organic matter (SOM); $f_{SA}$, $f_{SS}$ and $f_{SR}$ are the fractions of decomposed structural

litter-C that is transferred to active SOM pool, slow SOM pool and released to

atmosphere in forms of $CO_2$, respectively. As in the algorithms in CENTURY model

(Parton et al., 1988), here $f_{SA} = CUE_{d\_SA} \times (1\text{-}f_{lig})$, $f_{SS} = CUE_{d\_SS} \times f_{lig}$, $f_{SR}= 1\text{-}(f_{SA}+f_{SS})$,

where $f_{lig}$ is the lignin fraction (0–1, dimensionless) in the structural litter pool, and

$CUE_{d\_SA}$ and $CUE_{d\_SS}$ are the CUE of C transformation from structural litter pool to

active and slow SOM pool, respectively.





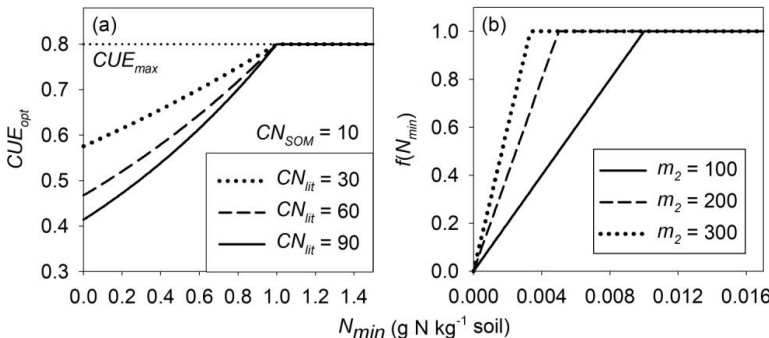


**Figure 2**. Schematic plot of (a) the optimal carbon use efficiency ($CUE_{opt}$) function of
soil mineral nitrogen for different litter C:N ratios (from Eq. (2) in the main text with
$m_1 = 0.3$, $n_1 = 1.0$) and (b) the N limitation function $f(N_{min})$ applied to litter
decomposition rates (from Eq. (5) in the main text). $CN_{lit}$ and $CN_{SOM}$ are the C:N ratio
of litter pool and SOM pool, respectively. $CUE_{max} = 0.8$ is the maximum $CUE$ under
optimal nutrient condition (C limitation only). $m_1$ and $n_1$ are the parameters of Eq. (3)
and $m_2$ are the parameter of Eq. (5).





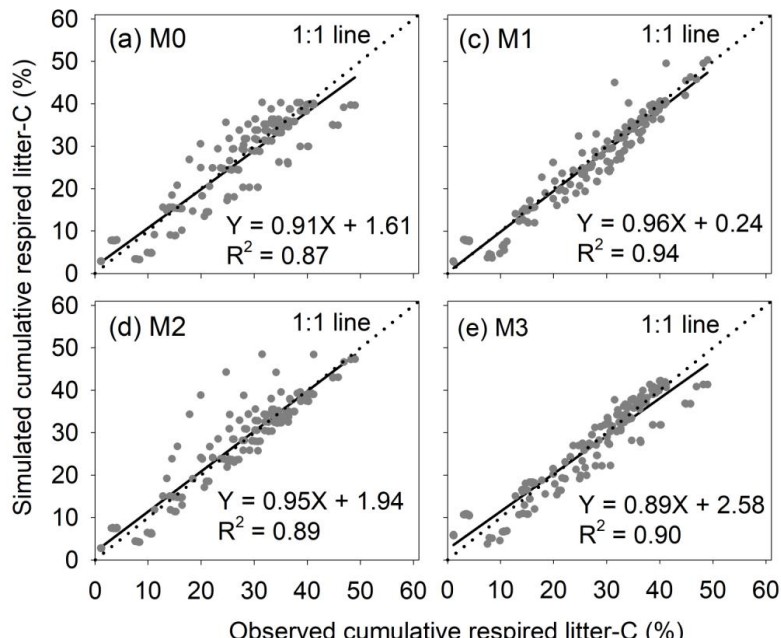


**Figure 3** Comparison of the predicted cumulative respired litter-C to observed values

at different times during litter decomposition process. Each dot denotes an

observation of cumulative respired litter-C at a certain day. Totally, there are 149

points. M0-M3 are the four versions of litter decay model tested in this study (Table

1).

847





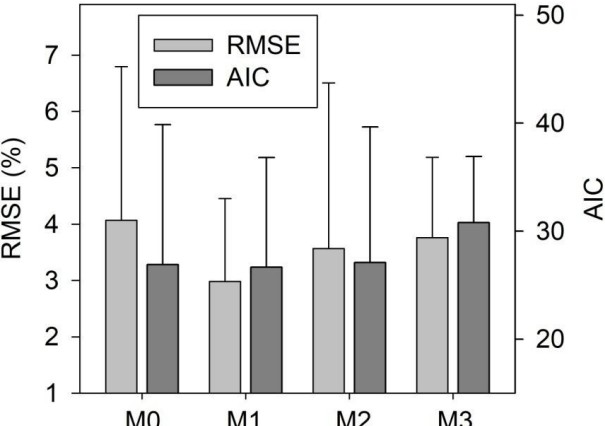

**Figure 4** The RMSE and AIC of the simulated cumulative respired litter-C from the four versions of litter decay model used in this study. Error bars denote the standard deviation of RMSE or AIC for different incubation experiments. M0 and M1-3 denote the four models tested in this study (Table 1).



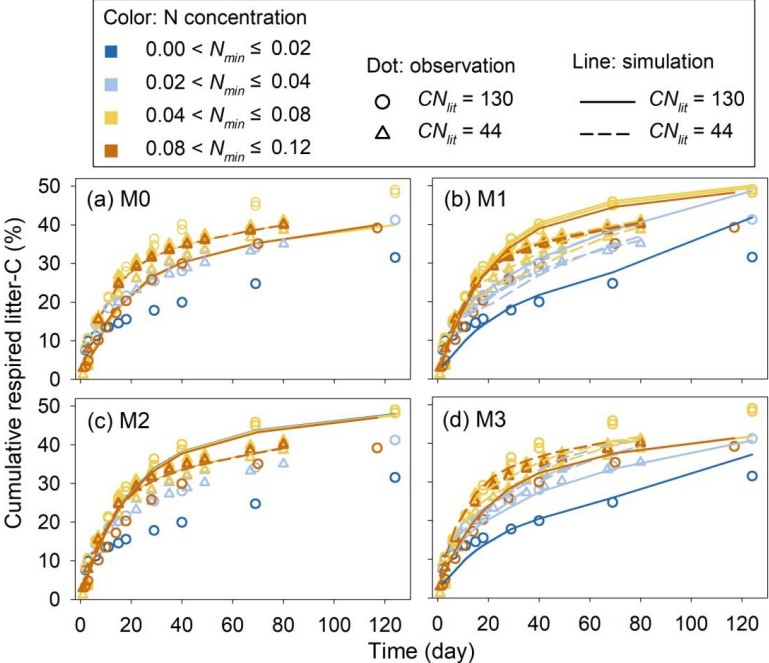

855

**Figure 5** Time series of the simulated (lines) and observed (dots) cumulative respired

litter-C (% of initial litter-C) at four different levels of soil mineral N availability ($N_{min}$,

g N kg$^{-1}$ soil). $CN_{lit}$ is the C:N ratio of plant litter. M0 and M1-3 denote the four

models tested in this study (Table 1). Here the simulation results of each model were

calculated with parameters optimized based on all of the 14 samples of incubation

experiments (Table A2).

862





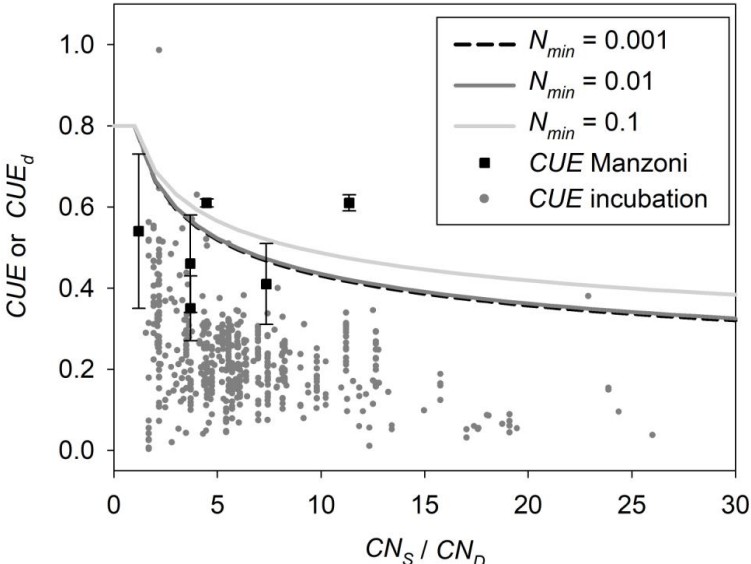

863

**Figure 6** Comparison of $CUE_d$ (lines) predicted by Eq. (2) with parameter values ($m_2$

= 0.54, $n_1$ = 0.50) calibrated based on the incubation experiments (Table A2) of

Recous et al. (1995) and Guenet et al. (2010) to observed $CUE$ of terrestrial

microorganisms along a gradient of $CN_S/CN_D$, where $CN_D$ and $CN_S$ are the C:N ratio

of decomposers and their substrates, respectively. Gray dots are the estimated

microbial CUE of litter decomposition in natural terrestrial ecosystems from Manzoni

*et al.* (2017). Black squares are the microbial CUE measured via laboratory

incubation experiments of Gilmour & Gilmour, (1985), Dev êvre & Horw áth (2000)

and Thiet *et al.* (2006). Error bars represent the standard deviations. $N_{min}$ (g N kg$^{-1}$ soil)

is the concentration of soil mineral N.

874



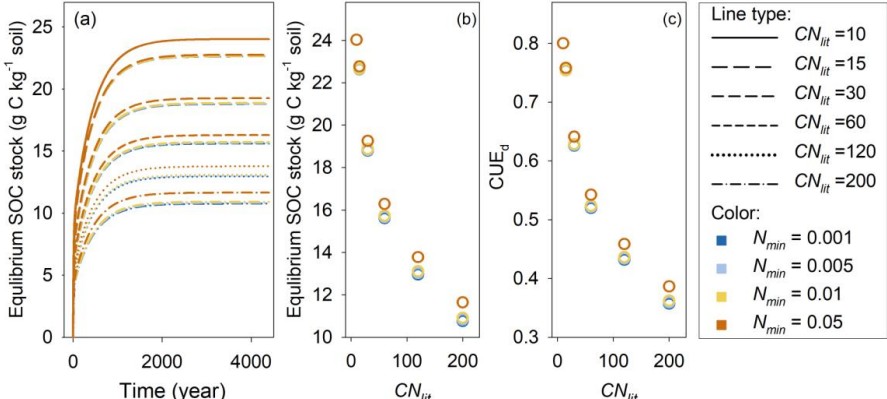

875

**Figure 7** (a) Accumulation of soil organic carbon (SOC) for constant substrates input

(plant litter) with different C:N ratios ($CN_{lit}$) at different levels of soil mineral N

concentrations ($N_{min}$, g N kg$^{-1}$ soil), (b) Change trends of equilibrium SOC stock and

carbon use efficiency of decomposed litter ($CUE_d$) with increasing litter C:N ratio.

880



## Appendix:

**Table A1** List of symbols used in this study

| Symbol | Unit | Description |
|---|---|---|
| $a$ | g N kg$^{-1}$ soil | Exponent in Eq. 2 |
| $AIC$ | dimensionless | The Akaike Information Criterion (Eq. 7) |
| $CN_{act}$ | dimensionless | C to N ratio of active soil organic matter pool |
| $CN_D$ | dimensionless | C to N ratio of decomposer (Eq. 1) |
| $CN_{met}$ | dimensionless | C to N ratio of metabolic litter pool |
| $CN_{slow}$ | dimensionless | C to N ratio of slow soil organic matter pool |
| $CN_{str}$ | dimensionless | C to N ratio of structural litter pool |
| $CN_S$ | dimensionless | C to N ratio of substrate (Eq. 1) |
| $C_{lit}$ | g C kg$^{-1}$ soil | C stock of litter pool (Eq. 4) |
| $CN_{lit}$ | dimensionless | C to N ratio of litter pool (metabolic or structural, Eq. 2) |
| $CN_{SOM}$ | dimensionless | C to N ratio of soil organic matter pool |
| $CUE$ | dimensionless | Microbial carbon use efficiency |
| $CUE_d$ | dimensionless | Carbon use efficiency of decomposition (C incorporated in SOC over litter C decomposed) |
| $CUE_{fit}$ | dimensionless | Optimized value of fixed CUE in model M0 and M4 |
| $CUE_{max}$ | dimensionless | Maximum $CUE_d$ (Eqs. 1 and 2) |
| $CUE_{opt}$ | dimensionless | Optimal $CUE_d$ (Eq. 1) |
| $CUE_{d\_SA}$ | dimensionless | CUE of the transformation from structural litter to active SOM pool |
| $CUE_{d\_SS}$ | dimensionless | CUE of the transformation from structural litter to slow SOM pool |
| $D$ | g C kg$^{-1}$ soil day$^{-1}$ | Daily litterfall input rate (Eq. 10) |
| $D(C_{lit-met})$ | g C kg$^{-1}$ soil day$^{-1}$ | Decomposition rate of metabolic litter |
| $D(C_{lit-str})$ | g C kg$^{-1}$ soil day$^{-1}$ | Decomposition rate of structural litter |
| $f(N_{min})$ | dimensionless | Limit factor of soil mineral N on litter decomposition (Eqs. 4 and 5) |
| $f(tem)$ | dimensionless | Limit factor of soil temperature on litter decomposition (Eq. 4) |
| $f(water)$ | dimensionless | Limit factor of soil water content on litter decomposition (Eq. 4) |
| $f_m$ | dimensionless | Fraction of metabolic plant litter |
| $f_{SA}$ | dimensionless | Fractions of decomposed structural litter-C that is transferred to active SOM pool |
| $f_{SR}$ | dimensionless | Fractions of decomposed structural litter-C that is released tp atmosphere |
| $f_{SS}$ | dimensionless | Fractions of decomposed structural litter-C that is transferred to slow SOM pool |
| $I_N$ | g kg$^{-1}$ soil | Maximum mineral N immobilization rate (Eq. 1) |
| $k$ | day$^{-1}$ | potential maximum turnover rate (Eq. 10) |
| $LC_{lit}$ | dimensionless | Lignin to C ratio of litter input |
| $m_1$ | kg g$^{-1}$ N | Coefficients in Eq. 3 |
| $n_1$ | g N kg$^{-1}$ soil | Coefficients in Eq. 3 |
| $m_2$ | day$^{-1}$ | Coefficients in Eq. 5 |
| $m_3$ | kg g$^{-1}$ N | Coefficients in Eq. 8 |
| $N_{min}$ | g N kg$^{-1}$ soil | Soil mineral N concentration (Eq. 5) |





| | | |
|---|---|---|
| *RMSE* | % | Root mean square error (Eq. 6) |
| *SOC* | g C kg$^{-1}$ soil | Soil organic carbon |
| *SOM* | g C kg$^{-1}$ soil | Soil organic matter |
| *U$_0$* | g C kg$^{-1}$ soil day$^{-1}$ | C uptake rate when soil mineral N is fully adequate for litter decay (Eq. 1) |

883



**Table A2** Information about the 14 samples of laboratory incubation experiment used in this study. $CN_{lit}$ and $LC_{lit}$ are the C to N ratio and lignin to C ratio of plant litter, respectively. $CN_{SOM}$ is the C to N ratio of SOM pool. $N_{min}$ is the concentration of soil mineral N ($NO_3^--N + NH_4^+-N$). For the incubation experiments of Guenet *et al.* (2010), cumulative respired litter-C was measured on days 1, 3, 7, 15, 22, 28, 35, 42, 49, 67 and 80, and $N_{min}$ was measured on days 3, 7, 17, 28 and 80. For For the incubation experiments of Recous *et al.* (1995), both cumulative respired litter-C and $N_{min}$ were mostly measured on days 2, 3, 6, 11, 15, 18, 29, 40, 69 and 124.

| Sample | $CN_{lit}$ | $LC_{lit}$ | $CN_{SOM}$ | Initial $N_{min}$ (g N kg⁻¹ soil) | Duration (day) | Temperature (℃) | Soil moisture (%, in volume) | Litter type | Reference |
|---|---|---|---|---|---|---|---|---|---|
| 1 | 44 | 0.26 | 11 | 0.035 | 80 | 20 | 50 | Crop (wheat) | |
| 2 | 44 | 0.26 | 11 | 0.051 | 80 | 20 | 50 | Crop (wheat) | |
| 3 | 44 | 0.26 | 11 | 0.055 | 80 | 20 | 50 | Crop (wheat) | |
| 4 | 44 | 0.26 | 11 | 0.033 | 80 | 20 | 50 | Crop (wheat) | Guenet *et al.*, 2010 |
| 5 | 44 | 0.26 | 11 | 0.049 | 80 | 20 | 50 | Crop (wheat) | |
| 6 | 44 | 0.26 | 11 | 0.067 | 80 | 20 | 50 | Crop (wheat) | |
| 7 | 44 | 0.26 | 11 | 0.033 | 80 | 20 | 50 | Crop (wheat) | |
| 8 | 44 | 0.26 | 11 | 0.048 | 80 | 20 | 50 | Crop (wheat) | |
| 9 | 44 | 0.26 | 11 | 0.079 | 80 | 20 | 50 | Crop (wheat) | |
| 10 | 130 | 0.23 | 9 | 0.010 | 124 | 15 | 42 | Crop (Corn) | |
| 11 | 130 | 0.23 | 9 | 0.030 | 124 | 15 | 42 | Crop (Corn) | Recous *et al.*, 1995 |
| 12 | 130 | 0.23 | 9 | 0.060 | 124 | 15 | 42 | Crop (Corn) | |
| 13 | 130 | 0.23 | 9 | 0.080 | 124 | 15 | 42 | Crop (Corn) | |
| 14 | 130 | 0.23 | 9 | 0.100 | 124 | 15 | 42 | Crop (Corn) | |



**Table A3** Specific setting of litter and SOM properties, and soil conditions in the 16
idealized simulations for exploring the impacts of litter stoichiometry (i.e. C:N ratio)
and soil mineral N on SOC accumulation. $CN_{lit}$ and $LC_{lit}$ are the C to N ratio and
lignin to C ratio of plant litter, respectively. $Lit_{inp}$ (g C kg$^{-1}$ soil day$^{-1}$) is the daily input
rate of plant litter. $CN_{SOM}$ is the C to N ratio of SOM pool. $N_{min}$ (g N kg$^{-1}$ soil) is the
concentration of soil mineral N ($NO_3^-$ -N + $NH_4^+$ -N). *Tem* (°C) and *SWC* (%) are the
temperature and soil water content, respectively.

| Experiment | $CN_{lit}$ | $LC_{lit}$ | $Lit_{inp}$ | $CN_{SOM}$ | $N_{min}$ | *Tem* | *SWC* |
|---|---|---|---|---|---|---|---|
| 1 | 15 | 0.2 | 0.006 | 12 | 0.001 | 25 | 60 |
| 2 | 30 | 0.2 | 0.006 | 12 | 0.005 | 25 | 60 |
| 3 | 60 | 0.2 | 0.006 | 12 | 0.01 | 25 | 60 |
| 4 | 120 | 0.2 | 0.006 | 12 | 0.05 | 25 | 60 |
| 5 | 15 | 0.2 | 0.006 | 12 | 0.001 | 25 | 60 |
| 6 | 30 | 0.2 | 0.006 | 12 | 0.005 | 25 | 60 |
| 7 | 60 | 0.2 | 0.006 | 12 | 0.01 | 25 | 60 |
| 8 | 120 | 0.2 | 0.006 | 12 | 0.05 | 25 | 60 |
| 9 | 15 | 0.2 | 0.006 | 12 | 0.001 | 25 | 60 |
| 10 | 30 | 0.2 | 0.006 | 12 | 0.005 | 25 | 60 |
| 11 | 60 | 0.2 | 0.006 | 12 | 0.01 | 25 | 60 |
| 12 | 120 | 0.2 | 0.006 | 12 | 0.05 | 25 | 60 |
| 13 | 15 | 0.2 | 0.006 | 12 | 0.001 | 25 | 60 |
| 14 | 30 | 0.2 | 0.006 | 12 | 0.005 | 25 | 60 |
| 15 | 60 | 0.2 | 0.006 | 12 | 0.01 | 25 | 60 |
| 16 | 120 | 0.2 | 0.006 | 12 | 0.05 | 25 | 60 |





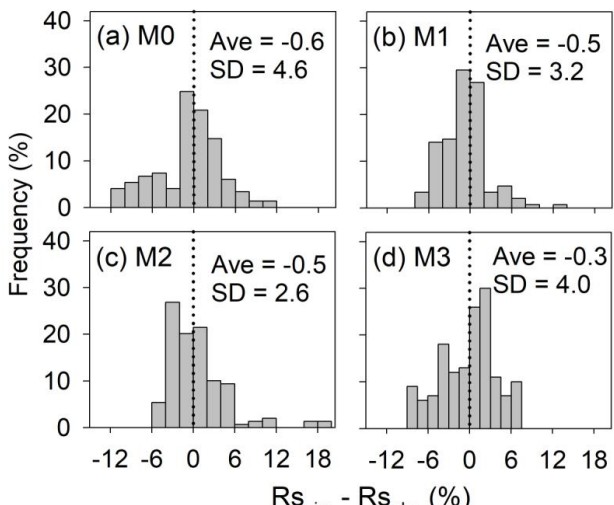

**Figure A1** Distribution of the difference between the predicted cumulative respired

litter-C ($Rs_{sim}$, %) and the observed values ($Rs_{obs}$, %) for all experiments and points in

time. SD is standard deviation of the biases. M0-M3 denote the four models tested in

this study (Table 1).

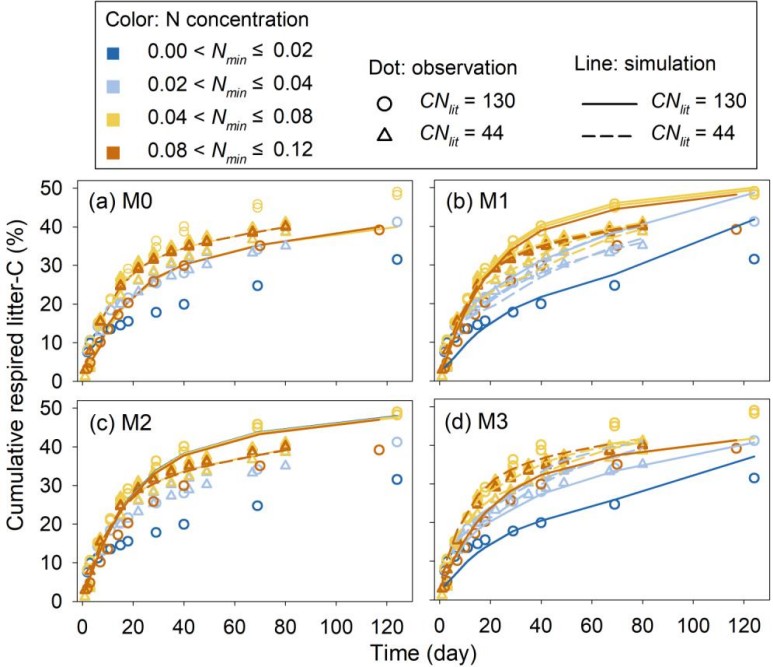

**Figure A2** Dynamic of the simulated carbon use efficiency ($CUE$) and $f(N_{min})$ during

the incubation experiments (Table S3). $CN_{lit}$ is the C:N ratio of incubated litter, and

$N_{min}$ is the initial soil mineral N concentration (g N kg$^{-1}$ soil). M0-M3 denote the four

models in Table 1. Here the simulation results of each model were calculated with

parameters optimized based on all of the 14 samples of incubation experiments (Table

S2).





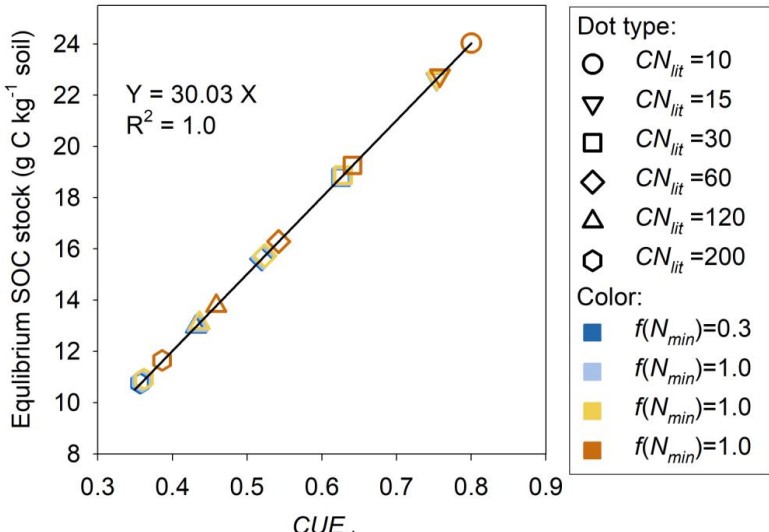

**Figure A3** Relationship between C stock of the potentially equilibrated SOM pool

and the carbon use efficiency of decomposed metabolic litter ($CUE_d$) at the dynamic

equilibrium stage. $f(N_{min})$ denote the inhibition factor (0–1) of soil mineral N on litter

decomposition.



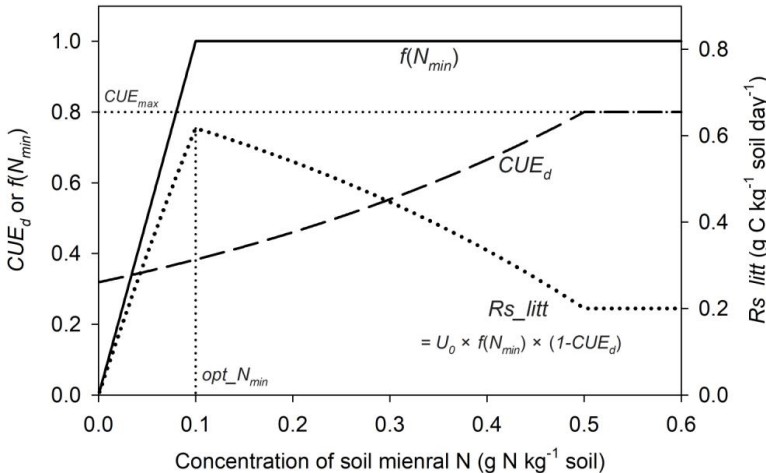

**Figure A4** Schematic plot for change trends of $f(N_{min})$ (inhibition effect of mineral N,
Eq. 6), $CUE_d$ (carbon use efficiency of decomposed litter, Eq. 2,3) and $Rs\_litt$ (litter
respiration rate) with increasing concentration of soil mineral N. $CUE_{max}$ (= 0.8) is the
maximum CUE set in this study. $opt\_N_{min}$ denotes the concentration of soil mineral N
at which litter respiration is maximized. $U_0$ is the potential decomposition rate when
mineral N is fully adequate for litter decay.





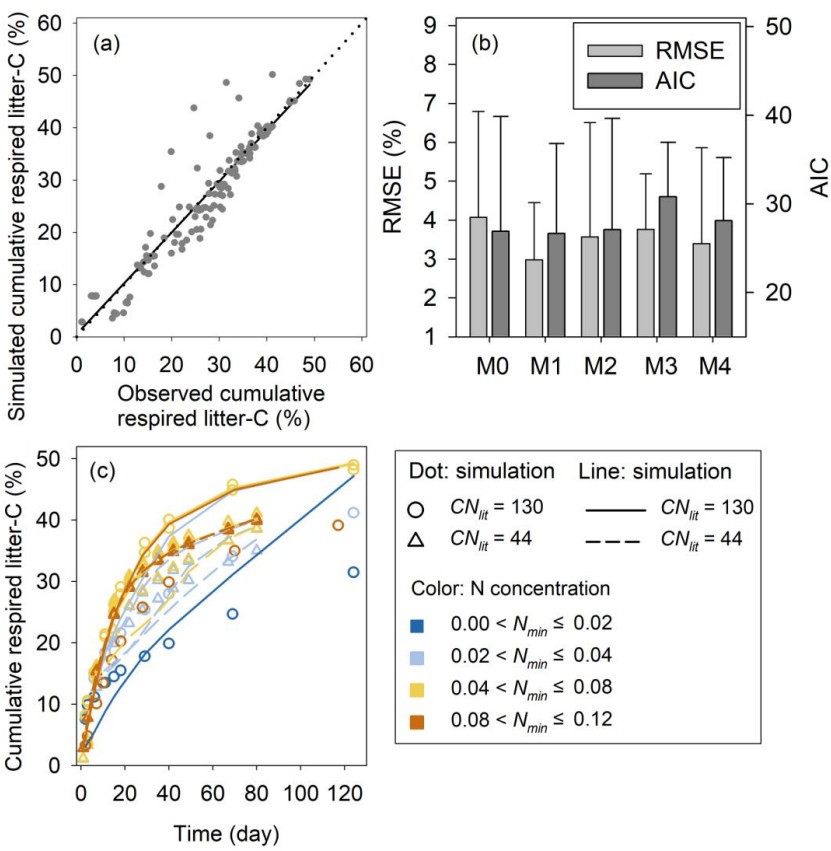

**Figure A5** Comparison between simulated cumulative respired litter-C with $f(N_{min})$

(inhibition effect of soil mineral N on litter decomposition) calculated by Eq. 9 and

the observed results from incubation experiments. In figure (c), M0-M3 denote the

four versions of litter decay model in Table 1. M4 denote the model which used Eq. 2

to calculate the dynamic CUE and used Eq. 9 to calculate $f(N_{min})$.

