# Peer review of "Modeling the effects of litter stoichiometry and soil mineral N"

_Geoscientific Model Development, 2018_

## Short Comment (SC1) · 15 Aug 2018

Dear authors,

in my role as Executive editor of GMD, I would like to bring to your attention our Editorial version 1.1:

http://www.geosci-model-dev.net/8/3487/2015/gmd-8-3487-2015.html

This highlights some requirements of papers published in GMD, which is also available on the GMD website in the 'Manuscript Types' section:

http://www.geoscientific-model-development.net/submission/manuscript_types.html

[Figure]

In particular, please note that for your paper, the following requirements have not been met in the Discussions paper:

- "The main paper must give the model name and version number (or other unique identifier) in the title."

- "If the model development relates to a single model then the model name and the version number must be included in the title of the paper. If the main intention of an article is to make a general (i.e. model independent) statement about the usefulness of a new development, but the usefulness is shown with the help of one specific model, the model name and version number must be stated in the title. The title could have a form such as, "Title outlining amazing generic advance: a case study with Model XXX (version Y)"."

In order to simplify reference to your developments, please add a model name (and/or its acronym) and a version number in the title of your article in your revised submission to GMD.

Yours,

Astrid Kerkweg

————————————————————

---

## Referee Comment (RC1) · Anonymous Referee #1 · 4 Sep 2018

General Comments: Zhang and coauthors present a numerically tractable way to introduce variable carbon use efficiency (CUE) into a first-order litter decomposition model based on nitrogen availability. The paper is well written, with a very clean introduction that nicely summarizes relevant literature and concludes with a clear organization of the paper. Methods are adequately descriptive, results are clearly presented, and the discussion is on target (but see comment on N enrichment and litter decay below).

Specific Comments: The approach outline here is nice, using short term experiments to calibrate the model and subsequently looking at the long-term dynamics. One concern, however, is that by using short term respiration rates from field and lab experiments to

[Figure]

calibrate the variable CUE it is not clear if turnover coefficients that control litter mass loss are at all appropriate (more on this below).

In section 2.5 it's a little unclear how the model and observations are disentangling background soil respiration from the litter respiration fluxes that are presumably being fit. Can this be clarified?

I'm assuming there are no modifications to other CUE terms in CENTURY (between SOM pools), but this should be clarified

Turnover times used in the model (e.g. tau_metabolic and tau_structural and well as the SOM turnover times listed in the github archive) are much larger than the litter turnover times used in CENTURY (Parton et al. 1988). This makes me wonder where the turnover parameterization here comes from? Addressing this concern is important since respiration rates are a product of turnover and CUE (given fixed initial pool sizes). Since the turnover times used here are much lower than in the CENTURY parameterization, the CUE will also have to be lower than if faster turnover times were used in the model. This is all fine, but should be made clearer in the text, which otherwise claims to be using the CENTURY approach.

The maximum CUE allowed in the study seems quite high (0.8, Fig. 2). I'm assuming this assumption also causes the apparent high bias in CUE shown in Fig. 6? Is the model able to fit the data as well with a more reasonable upper limit for CUE (say 0.6), or is the high efficiency needed to capture results observed in the experiment?

The main response of changes in CUE with litter quality seem to be opposite of what's expected. It seems like the authors expected to see a "decrease in CUEd with decreasing litter quality" (line 224), but instead report higher CUEd with the lower quality litter (line 363). Please explain how the parameterization let to this response and seems to contradict findings reported in Fig. 6.

Line 400. I agree, it's nice these parameters can be estimated, but the fit parameter

values and their associated uncertainty are never communicated in the text. Can they be given in Table 1, or elsewhere in the manuscript? Similarly, does it make sense to include parameter values in Table A1?

The discussion is largely on target and I was very excited to see the authors try to take on results that generally show lower litter decomposition rates with N enrichment (e.g. Fog 1988, Knorr et al. 2005), line 415. What follows, however, does not really conceptually address the apparent paradox of N additions, litter decay, and CUE. Instead the mathematical approach introduces new simulations and a new set of simulations (eq. 8, 9 & Fig. A5). Introducing new results like this in the discussion seems inappropriate for the journal. Instead it seems like these findings could be: (a) incorporated into the method and results; or (b) dropped from the manuscript. I would encourage the first option, but also ask the authors to more thoughtfully discuss how their results can inform larger questions about litter decay and N enrichment (Nave et al. 2009; Hobbie 2015; see also Wieder et al. 2015).

My final concern is somewhat subjective, but I argue that litter decomposition and SOM formation are not the same process. Throughout, however, the text (and especially the discussion) misleadingly conflates these two processes. While it's true that in first order models like CENTURY these processes are intimately linked, a growing body of literature highlights fundamental differences between processes controlling litter decay and SOM formation (see Lehmann and Kleber 2015, Sokol et al. 2018). Results shown in Fig. 7 are fine, but I would caution against linking these processes directly in the text.

Technical corrections: Line 215, Don't 'microbes' include fungi and bacteria?

Line 215, Cleveland and Liptzin report microbial C:N = 8.6 (molar), so I'm assuming the 7.4 reported here on a mass basis, but this should be clarified in the text?

Methods: It may be helpful to describe how the model handles partitioning of litter into metabolic and structural litter pools, and how the stoichiometry of these LIT pools changes with changes in litter quality (e.g., what are the donor pool C:N ratios if litterfall

inputs have a C:N of 40 vs. 130)? .

Line 270 & 319, seems odd to cite a web site for a corporation selling composting material. A better choice may be Brovkin et al. 2012, who report litter quality estimates from the ART-DECO database, or work from the LIDET team (e.g. Harmon et al. 2009).

Line 355, this statement isn't very obvious from Fig. 5b, in my estimation.

Fig. 6. It's not really clear how the authors plot the C:N ratio of substrates : decomposers for a model that doesn't consider decomposers. I'm assuming this is the C:N ratio of donor (litter) / receiver (SOM pools; eq. 2)? Maybe this can be clarified in the figure caption? This is a fine assumption to make, although Cleveland and Liptzin (2007) found microbial C:N < soil C:N.

How is Fig. A2 different from Fig. 5? Moreover, the caption in A2 doesn't seem to match the display item? (see also lines 374, 376).

From line 480-506 on the discussion wanders well beyond the scope of results presented here. In particular, the emphasis on humic substances and litterfall driving SOM formation seems well out of line with contemporary thinking about factors controlling SOM stabilization (Lehmann and Kleber 2015). Moreover, the positive connection between CUE and steady-state SOM pools in first order models is well established (e.g. Frey et al. 2013). What's nice with the work presetned here is the ability to link ideas about litter quality and SOM formation in ways that are consistent with theory about CUE and substrate quality (MEMs conceptual model, Cortufo et al. 2013) in a first order model. I'd encourage the authors more closely stick to interpreting the results presented with this work.

Line 512, self-citations are nice, but it may also be worth referencing other modeling groups here?

Line 516, didn't Bonan and others (2013, cited elsewhere in the text) already do this with CLM and CENTURY? Seems worth crediting work that's already been done along

these lines.

Line 538, the comparison with 'most large-scale litter decay models' was not made in this paper and I would remove this clause from the conclusion.

References:

Brovkin, V., van Bodegom, P. M., Kleinen, T., Wirth, C., Cornwell, W., Cornelissen, J. H. C., & Kattge, J. (2012). Plant-driven variation in decomposition rates improves projections of global litter stock distribution. Biogeosciences, 9, 565-576. doi: 10.5194/bg-9-565-2012.

Fog, K. (1988). The effect of added nitrogen on the rate of decomposition of organic matter. Biol. Rev., 63, 433-462.

Frey, S. D., Lee, J., Melillo, J. M., & Six, J. (2013). The temperature response of soil microbial efficiency and its feedback to climate. Nature Clim. Change, 3, 395-398. doi: 10.1038/nclimate1796.

Harmon, M. E., Silver, W. L., Fasth, B., Chen, H. U. A., Burke, I. C., Parton, W. J., et al. (2009). Long-term patterns of mass loss during the decomposition of leaf and fine root litter: an intersite comparison. Global Change Biology, 15(5), 1320-1338. doi: 10.1111/j.1365-2486.2008.01837.x.

Hobbie, S. E. (2015). Plant species effects on nutrient cycling: revisiting litter feedbacks. Trends Ecol Evol, 30(6), 357-363. doi: 10.1016/j.tree.2015.03.015.

Knorr, M., Frey, S. D., & Curtis, P. S. (2005). Nitrogen additions and litter decomposition: A meta-analysis. Ecology, 86(12), 3252-3257. doi: 10.1890/05-0150.

Lehmann, J., & Kleber, M. (2015). The contentious nature of soil organic matter. Nature, 528(7580), 60-68. doi: 10.1038/nature16069.

Nave et al. (2009) Impacts of elevated N inputs on north temperate forest soil C storage, C/N, and net N-mineralization. Geoderma 153, 231–240.

Sokol, et al. (2018) "Evidence for the primacy of living root inputs, not root or shoot litter, in forming soil organic carbon." New Phytologist doi: 10.1111/nph.15361.

Wieder, W. R., Grandy, A. S., Kallenbach, C. M., Taylor, P. G., & Bonan, G. B. (2015). Representing life in the Earth system with soil microbial functional traits in the MIMICS model. Geoscientific Model Development, 8(6), 1789-1808. doi: 10.5194/gmd-8-1789-2015.

---

## Referee Comment (RC2) · Anonymous Referee #2 · 8 Sep 2018

This study adapted a conceptual formulation of CUEd based on assumption that litter decomposers optimally adjust their CUEd as a function of litter substrate C to nitrogen (N) stoichiometry. The new model algorithm was incorporated into CENTURY soil biogeochemical model and evaluated using data from laboratory litter incubation experiments. The results showed that new CUEd formulation with flexible CUE and effect of N availability to decay rate was able to reproduce differences in respiration rate of litter with contrasting C:N ratios and under different levels of mineral N availability. It is well-written, logically organized, and the figures and tables are appropriate.

Figure 1 seems too simple to include other major processes mentioned in the method

[Figure]

section. It should be considered to revise.

As the CUEd was defined as a fraction of it is respired to the atmosphere and the remaining fraction (Line 159-160), it is not correct to use 1-CUEd to simulate CO2 emission in Fig. 1.

Equ (4) is important for this study, which has been used to develop one of model simulations (i.e. M1). However what is the fundamental assumption for adding N effects in the Equ (4)? N mineralization is accompanied with carbon decomposition. So, why use N availability to limit litter decay?

Line 71: need reference here.

Line 212: typo "The The C:N ratio"

---

## Author Comment (AC1) · 2 Oct 2018

Dear editor,

We received the comments from the executive editor and the two referees on our manuscript "Modeling the effects of litter stoichiometry and soil mineral N availability on soil organic matter formation" (gmd-2018-173). We are very grateful for their constructive comments and suggested amendments. We have carefully studied them, and revised our manuscript accordingly. As a consequence, our manuscript has been considerably improved.

The following part is our detailed responses to the comments from the executive editor and referees. Please note that the comments are in **bold** followed by our responses in regular text.

Sincerely,

Haicheng Zhang, on hehalf of all coauthors
Email: haicheng.zhang@lsce.ipsl.fr

**Response to the Executive editor of GMD**

**1. In my role as Executive editor of GMD, I would like to bring to your attention our Editorial version 1.1: http://www.geosci-model-dev.net/8/3487/2015/gmd-8-3487-2015.html. This highlights some requirements of papers published in GMD, which is also available on the GMD website in the 'Manuscript Types' section: http://www.geoscientific-model-development.net/submission/manuscript_types. html.**

Thank you for this reminder. We have read the requirements of paper published in GMD carefully, and also adapted our manuscript accordingly to ensure it meets all the requirements of GMD. See below for details.

**2. In particular, please note that for your paper, the following requirements have not been met in the Discussions paper: • "The main paper must give the model name and version number (or other unique identifier) in the title." • "If the model development relates to a single model then the model name and the version number must be included in the title of the paper. If the main intention of an article is to make a general (i.e. model independent) statement about the usefulness of a new development, but the usefulness is shown with the help of one specific model, the model name and version number must be stated in the title. The title could have a form such as, "Title outlining amazing generic advance: a case study with Model XXX (version Y)". In order to simplify reference to your developments, please add a model name (and/or its acronym) and a version number in the title of your article in your revised submission to GMD.**

To fulfill these requirements, we have added the model name and version number in the title of our article. The original title has been changed from "Modeling the effects of litter stoichiometry and soil mineral N availability on soil organic matter formation"

to

"Modeling the effects of litter stoichiometry and soil mineral N availability on soil organic matter formation using CENTURY-CUE (v1.0)". (see lines 1-3)

**Response to Referee #1**

**1.General Comments: Zhang and coauthors present a numerically tractable way to introduce variable carbon use efficiency (CUE) into a first-order litter decomposition model based on nitrogen availability. The paper is well written, with a very clean introduction that nicely summarizes relevant literature and concludes with a clear organization of the paper. Methods are adequately descriptive, results are clearly presented, and the discussion is on target (but see comment on N enrichment and litter decay below).**

Thank you for your positive comments, and please see our responses to your concerns below.

**2. Specific Comments: The approach outline here is nice, using short term experiments to calibrate the model and subsequently looking at the long-term dynamics. One concern, however, is that by using short term respiration rates from field and lab experiments to calibrate the variable CUE it is not clear if turnover coefficients that control litter mass loss are at all appropriate (more on this below).**

Indeed, the litter turnover times have significant impacts on the fitted values of CUE. In our study, the turnover times for C pools are obtained from the ORCHIDEE-MICT that has good performances in reproducing observed organic carbon pools (v8.4.1, Guimberteau *et al*., 2018). However, we have calibrated the turnover times of the litter pools to the data of the incubation experiments. This calibration was necessary because the plant residues used in the incubation experiments of Recous *et al*. (1995) and Guenet *et al*. (2010) had been cut into fine fragments before being mixed with soil. It is known that the decomposability of litter is negatively correlated to its physical size (Tuomi *et al*., 2011). We further argue that the mixing increases the accessibility of litter for microbes. Therefore, the turnover times of the incubated litter used in the experiments of Recous *et al*. (1995) and Guenet *et al*. (2010) should be shorter than the litter turnover times set in ORCHIDEE-MICT (24 days for metabolic litter and 89 days for structural litter), which are representative of to the turnover times of natural plant residues. In this study, we calibrated the turnover times of litter pools (metabolic and structural) based on the observed cumulative respired litter-C from all of the 14 incubation experiments using the M0 and M1 model (see Table A3 below).

We have added one paragraph to introduce the source of the SOC turnover times used in this study, and how we have calibrated the litter turnover times: "Note that the turnover times of SOM pools (active, slow and passive) used in this study are obtained from the ORCHIDEE-MICT (v8.4.1, Guimberteau *et al*., 2018). The turnover times of litter pools (metabolic and structural), as well as the coefficient $m_4$ in Eq. (8) were optimized against the observed cumulative respired litter-C from all of the 14 incubation experiments using the M0 and M1 models (Table A3). A previous study has shown that litter decomposability is negatively correlated to its physical size (for example, Tuomi *et al*., 2011). Therefore, the turnover times of the fine litter fragments used in the incubation experiments of Recous *et al*. (1995) and Guenet *et al*. (2010) are expected to be shorter than the values set in ORCHIDEE-MICT, which are representative of the turnover times of natural plant residues. In addition, the mixing of soil and litter particle in the incubation experiment likely enhances decomposition as spatial disconnection of decomposer and substrate, which can occur under natural soil conditions (Barnes *et al*., 2012; Hewins *et al*., 2013), is prevented. The calibrated turnover times of the metabolic and structural pools and the value of $m_4$ in Eq. (8) are 3.5 and 30 days and 0.5, respectively." (lines 465-483)

**Table A3** List of parameters calibrated for two versions of the litter decomposition model (M0, M1): $k_{litm}$ and $k_{lits}$ are respectively the turnover rates of metabolic and structural litter pools, $m_4$ is the coefficient in Eq. (8), $cue_{fit}$ is the optimized value of

CUE, $m_1$ and $n_1$ are the coefficients in Eq. (3), and $m_2$ is the coefficients in Eq. (5).

| Version | CUE | $f(N_{min})$ | Parameters |
|---------|-----|-----------|------------|
| M0 | Fixed | 1 | $cue_{fit}$, $k_{litm}$, $k_{lits}$, $m_4$ |
| M1 | Eqs.(2), (3) | Eq. (5) | $m_1$, $n_1$, $m_2$, $k_{litm}$, $k_{lits}$, $m_4$ |

(lines 1532-1536)

**3. In section 2.5 it's a little unclear how the model and observations are disentangling background soil respiration from the litter respiration fluxes that are presumably being fit. Can this be clarified?**

We have added some sentences to explain how the model and observations distinguish the litter- and SOC-derived $CO_2$.

For incubation experiments:

"To distinguish the litter- and SOC-derived $CO_2$ flux, Guenet *et al*. (2010) used straw from wheat grown under $^{13}C$ labeled $CO_2$ and they are therefore able to track the $CO_2$ coming from litter and the $CO_2$ coming from soil. In the experiments by Recous et al. (1995), litter-derived $CO_2$ flux is calculated as the difference in $CO_2$ flux between the incubation samples with both soil and litter, and the control samples without added litter." (lines 372-377)

For simulations:

"The observed cumulative respired litter-C (g C $kg^{-1}$ soil) measured in the incubation experiments was used to calibrate the model parameter values. Moreover, to quantify the simulated $CO_2$ flux derived from the litter, we also performed a set of control simulations with only SOM (initial litter pools were set to 0 g $kg^{-1}$ soil) using the four model versions. The simulated litter-derived $CO_2$ flux is calculated as the difference in $CO_2$ flux between the simulation with both litter and SOM inputs and the simulation with only SOM input." (lines 408-415)

**4. I'm assuming there are no modifications to other CUE terms in CENTURY (between SOM pools), but this should be clarified.**

We have added some sentences to clarify that only CUE for C transfers from litter pools to SOC pools were modified. Please see:

"Eqs. (2) and (3) were implemented in CENTURY to modify the originally fixed $CUE_d$ (Fig. 1). With this change, the fractions of C from litter that remain in SOM are mediated by stoichiometric constraints and mineral N availability, at the expense of additional parameters to fit. The $CUE_d$ for C transfers between SOC pools (active, slow and passive) are not modified." (lines 295-299)

**5. Turnover times used in the model (e.g. tau_metabolic and tau_structural and well as the SOM turnover times listed in the github archive) are much larger than the litter turnover times used in CENTURY (Parton et al. 1988). This makes me wonder where the turnover parameterization here comes from? Addressing this concern is important since respiration rates are a product of turnover and CUE (given fixed initial pool sizes). Since the turnover times used here are much lower than in the CENTURY parameterization, the CUE will also have to be lower than if faster turnover times were used in the model. This is all fine, but should be made clearer in the text, which otherwise claims to be using the CENTURY approach.**

The reviewer is correct; please see our response to Comment #2.

**6. The maximum CUE allowed in the study seems quite high (0.8, Fig. 2). I'm assuming this assumption also causes the apparent high bias in CUE shown in Fig. 6? Is the model able to fit the data as well with a more reasonable upper limit for CUE (say 0.6), or is the high efficiency needed to capture results observed in the experiment?**

We agree that CUE=0.8 is a relatively high value. While the CUEs of soil microbes are mostly concentrated between 0.4 and 0.6 (Manzoni et al., 2012), maximum values for reduced substrates are around 0.8 (Gommers et al., 1988), similar to maximum values also found in soils (Manzoni et al., 2012). Therefore, to allow the calibration procedure to cover a wide range of microbial CUEs, we set the maximum

CUE to 0.8. We have indicated the source reference of the maximum CUE in our manuscript. Please see:

"$CUE_{max}$ (dimensionless) is the maximum $CUE_d$ achieved when nutrients are not limiting, and it is set to 0.8 based on a synthesis of observed CUE of soil microbes (Manzoni *et al*., 2012)." (lines 264-266)

In addition, we also tested the performance of M1 model using a lower $CUE_{max}$ of 0.6 as the referee suggests to be more reasonable. The result indicates that the optimized M1r is also able to capture the differences in respiration rates due to different C:N ratios of substrate and varying levels of mineral N availability across the 14 incubation experiments (Fig. R1b), although the RMSE (also AIC) of its estimation is slightly higher than that of M1 (Fig. R1c). The optimized function of *f($N_{min}$)* (Eq. 5) with a $CUE_{max}$ of 0.6 is almost same to that with a $CUE_{max}$ of 0.8 (Fig. R2b). But the optimized $CUE_d$ function (Eq. 2) with a $CUE_{max}$ of 0.6 is very different from that with a $CUE_{max}$ of 0.8. When the $CUE_{max}$ is set to 0.6, $CUE_d$ increases very slowly with increasing soil mineral N concentration (Fig R2a), and shows almost no difference for litter with different qualities.

[Figure]

**Figure R1** Comparison of simulated cumulative respired litter-C between models with CUE upper limit of 0.8 (M1) and 0.6 (M1r), respectively.

[Figure]

**Figure R2**. Change in the relations between carbon use efficiency ($CUE_d$, (a)) and N limitation factor ($f(N_{min})$, (b)), and mineral N concentration. Here the $CUE_d$ and $f(N_{min})$ are calculated based on the optimized parameters when the upper limit of CUE is set to 0.8 (continuous line) and 0.6 (dashed line), respectively. $CN_{lit}$ and $CN_{SOM}$ are the C:N ratios of litter and SOM pools, respectively.

**7. The main response of changes in CUE with litter quality seem to be opposite of what's expected. It seems like the authors expected to see a "decrease in CUEd with decreasing litter quality" (line 224), but instead report higher CUEd with the lower quality litter (line 363). Please explain how the parameterization let to this response and seems to contradict findings reported in Fig. 6.**

There was a mistake in the text. We found an increase in $CUE_d$ with declining litter quality. We revised the text:

"For very low quality litter with a C:N ratio of 130, the $CUE_d$ in models M1 and M2 are 0.55 and 0.56, respectively, which are higher than for better quality litter with C:N ratio of 44 (approximately 0.40 and 0.44 in M1 and M2, respectively)."

to

"For very low quality litter with a C:N ratio of 130, the $CUE_d$ in models M1 and M2 are 0.40 and 0.44, respectively, which are lower than for better quality litter with C:N ratio of 44 (approximately 0.55 and 0.56 in M1 and M2, respectively)." (lines 616-640)

**8. Line 400. I agree, it's nice these parameters can be estimated, but the fit parameter values and their associated uncertainty are never communicated in the text. Can they be given in Table 1, or elsewhere in the manuscript? Similarly, does it make sense to include parameter values in Table A1?**

We have added the parameter values and their associated uncertainties to the Table 1 in our manuscript. Please see:

"**Table 1** Optimized parameter values for the five versions of the litter decomposition model used in this study. $cue_{fit}$ is the optimized value of CUE, $m_1$ and $n_1$ are the coefficients in Eq. (3), $m_2$ is the coefficient in Eq. (5), and $m_3$ is the coefficient in Eq. (6). Values in brackets following each parameter are the means ($\pm$ standard deviations) of the fitted parameter values based on "leave-one-out" cross-validation (see Section 2.5 for more details).

| Version | CUE | $f(N_{min})$ | Parameters |
|---------|-----|--------------|------------|
| M0 | Fixed | 1 | $cue_{fit}$ (0.57±0.004) |
| M1 | Eqs. (2), (3) | Eq. (5) | $m_1$ (0.61±0.34), $n_1$ (0.53±0.21), $m_2$ (297.4±38.0) |
| M2 | Eqs. (2), (3) | 1 | $m_1$ (0.11±0.01), $n_1$ (1.96±0.13) |
| M3 | Fixed | Eq. (5) | $cue_{fit}$ (0.54±0.01), $m_2$ (396.9±23.6) |
| M4 | Eqs.(2), (3) | Eq. (6) | $m_1$ (0.13±0.07), $n_1$ (1.91±0.37), $m_3$ (0.58±0.12) |

" (lines 1409-1416)

**9. The discussion is largely on target and I was very excited to see the authors try to take on results that generally show lower litter decomposition rates with N enrichment (e.g. Fog 1988, Knorr et al. 2005), line 415. What follows, however,**

**does not really conceptually address the apparent paradox of N additions, litter decay, and CUE. Instead the mathematical approach introduces new simulations and a new set of simulations (eq. 8, 9 & Fig. A5). Introducing new results like this in the discussion seems inappropriate for the journal. Instead it seems like these findings could be: (a) incorporated into the method and results; or (b) dropped from the manuscript. I would encourage the first option, but also ask the authors to more thoughtfully discuss how their results can inform larger questions about litter decay and N enrichment (Nave et al. 2009; Hobbie 2015; see also Wieder et al. 2015).**

Thanks for your suggestion. We have moved the description of the alternative formulation for $f(N_{min})$ from the discussion section to the method section(see Section 2.4, lines: 326-340).

We added:

"The Model M4, which uses the alternative formulation for N constraints on litter decay (Eq. (6)), reproduces the different respiration rates of substrates with contrasting C:N ratios and at different levels of mineral N availability (Fig. A2), but with a slightly higher average RMSE of cumulative respired litter-C than model M1."in the Results section (lines 605-609)

We also added:

"In addition, the model M4, which is comparable to model M1 but uses an alternative formulation for N effects on the decomposition rate (Eq. (6)), performed slightly worse than model M1 (Fig. A2). Arguably, Eq. (6) represents the underlying mechanisms of N inhibition effects (Manzoni *et al.*, 2009; Bonan *et al.*, 2013; Fujita *et al.*, 2014; Averill and Waring, 2018) better than Eq. (5) and due to the minor differences in RMSE and AIC (Figure A2b) between these formulations it can serve as an alternative to M1." in the Discussion section (lines 701-707)

Moreover, we have revised the discussions on the effects of N enrichment on litter respiration rate. The original sentences have been changed from "Moreover, describing N limitations on both the decomposition rate and flexible CUE$_d$ might allow our model to explain the observed diverse responses of litter respiration rate to added mineral N in fertilization experiments (Hobbie and Vitousek, 2000; Guenet *et al.*, 2010; Janssens *et al.*, 2010). In these experiments, the net changes in respiration rate depend on the combined effects of added N on litter decay rate and $CUE_d$ of the decayed litter (Fig. A4)."

to

"Our results indicate that the observed diversity of responses of litter respiration rate to mineral N additions (Hobbie and Vitousek, 2000; Guenet *et al.*, 2010; Janssens *et al.*, 2010) is likely due to the combined effects of changes in litter decay rate and $CUE_d$ (Fig. A5). Thus, N addition effects can differ among fertilization experiments if litter quality and background N availability vary. In addition to altering litter decay rate and $CUE_d$, mineral N addition can induce abiotic formation of compounds that resist microbial attack, inhibit oxidative enzymes involved in lignin degradation, stimulate microbial biomass production early in decomposition, or lead to the accumulation of microbial residues that are resistant to decay (Fog, 1988; Hobbie, 2015). All these effects might decrease litter respiration rate by inhibiting the decomposition process, but have not been considered in our current model." (lines 708-728)

**10. My final concern is somewhat subjective, but I argue that litter decomposition and SOM formation are not the same process. Throughout, however, the text (and especially the discussion) misleadingly conflates these two processes. While it's true that in first order models like CENTURY these processes are intimately linked, a growing body of literature highlights fundamental differences between processes controlling litter decay and SOM formation (see Lehmann and Kleber 2015, Sokol et al. 2018). Results shown in Fig. 7 are fine, but I would caution against linking these processes directly in the text.**

This is also a good point. Indeed, litter decomposition and SOM formation are not the same processes, since SOM formation also involves stabilization processes. However, the first-order decomposition models like CENTURY have represented these complicated processes in a very simple way, without explicit representation of the continuous transformation processes from decomposed litter to microbial productions and finally to stable SOM. According to your suggestion, we have revised our manuscript and deleted the sentences which might misleadingly conflate the litter decomposition and SOM formation processes. The major revision can be found from our response to your Comment #17 below. Please see lines 729-740 of the revised manuscript.

**11. Technical corrections: Line 215, Don't 'microbes' include fungi and bacteria?**
**Line 215, Cleveland and Liptzin report microbial C:N = 8.6 (molar), so I'm assuming the 7.4 reported here on a mass basis, but this should be clarified in the text?**

We have changed the original sentences from

"The C:N ratio of SOM (around 9:1 on a mass basis in CENTURY) is representative of the decomposer biomass, its value being between the C:N ratios of the two major group decomposers, soil microbes (7.4:1) (Cleveland and Liptzin, 2007)and soil fungi (13.4:1, Zhang *and* Elser, 2017)."

to

"The C:N ratio of SOM (around 9:1 on a mass basis in CENTURY) is representative of the decomposer biomass, its value being between the average C:N ratio of soil microbial communities including fungi and bacteria (7.4:1 in Cleveland and Liptzin, 2007) and the C:N ratio of soil fungi (13.4:1 in Zhang and Elser, 2017), which are probably largely responsible for fresh litter decomposition.". (lines 259-264)

**12. Methods: It may be helpful to describe how the model handles partitioning of litter into metabolic and structural litter pools, and how the stoichiometry of these LIT pools changes with changes in litter quality (e.g., what are the donor pool C:N ratios if litterfall inputs have a C:N of 40 vs. 130)?**

We have added a few sentences to introduce how the litter input is partitioned into metabolic and structural pools, as well as how we set the C:N ratio of litter pools. Please see:

"Plant litter was firstly separated into metabolic and structural litter pools based on its lignin to C ratio ($LC_{lit}$, dimensionless). The fraction of metabolic litter-C ($f_m$, 0-1, dimensionless) is calculated by:

$$f_m = f_{max} - m_4 \times LC_{lit} \tag{8}$$

where $m_4$ is a coefficient to be calibrated; $f_{max}$=0.85 is the maximum fraction of metabolic litter (i.e., the default value in CENTURY; Parton $et\ al.$, 1988). The fraction of structural litter-C is thus 1- $f_m$. The C:N ratios of both metabolic and structural pools are assumed to be equal to the C:N ratio of litter input." (lines 398-406)

Note that, to avoid a double-consideration of the N content of litter input (that is to say the C:N ratio has been involved in the CUE formula), we just use the lignin content (Lignin:C) to calculate the fraction of metabolic litter. This is different from the algorithm used in the default CENTURY, which separates the litter inputs into metabolic and structural pools based on both lignin and N content.

**13. Line 270 & 319, seems odd to cite a web site for a corporation selling composting material. A better choice may be Brovkin et al. 2012, who report litter quality estimates from the ART-DECO database, or work from the LIDET team (e.g. Harmon et al. 2009).**

Thanks for your suggestion. We have changed the original sentence from "The C:N ratios of those corn residue and wheat straw span the range of litter C:N ratios among different ecosystems (Harmon $et\ al.$, 2009; https://www.planetnatural.com/composting-101/making/c-n-ratio/)."

to

"The C:N ratios of those corn residue and wheat straw span the range of litter C:N ratios among different ecosystems (Harmon $et\ al.$, 2009; Brovkin $et\ al.$, 2012; Manzoni $et\ al.$, 2010)." (lines 365-367)

and from "The assumed litter C:N ratios ($CN_{lit}$) of 10, 15, 30, 60, 120 and 200 span the variation among most natural substrates and soil amendments from organic matter input in agriculture (Manzoni *et al*., 2012; https://www.planetnatural.com/composting-101/making/c-n-ratio/)."

to

"The assumed litter C:N ratios ($CN_{lit}$) of 10, 15, 30, 60, 120 and 200 span the variation among most natural substrates and soil amendments from organic matter input in agriculture (Harmon *et al*., 2009; Brovkin *et al*., 2012; Manzoni *et al*., 2010)." (lines 490-493)

**14. Line 355, this statement isn't very obvious from Fig. 5b, in my estimation.**

We have deleted the sentence "In addition, model M1 can also capture the temporal evolution of cumulative respired litter-C in different incubation experiments (Fig. 5b)."

**15. Fig. 6. It's not really clear how the authors plot the C:N ratio of substrates : decomposers for a model that doesn't consider decomposers. I'm assuming this is the C:N ratio of donor (litter) / receiver (SOM pools; eq. 2)? Maybe this can be clarified in the figure caption? This is a fine assumption to make, although Cleveland and Liptzin (2007) found microbial C:N < soil C:N.**

Sorry for the unclear explanation on the x-axis of Fig. 6. We have changed the original figure caption from "**Figure 6** Comparison of $CUE_d$ (lines) predicted by Eq. (2) with parameter values (m2 = 0.54, n1 = 0.50) calibrated based on the incubation experiments (Table A2) of Recous *et al*. (1995) and Guenet *et al*. (2010) to observed CUE of terrestrial microorganisms along a gradient of $CN_S/CN_D$, where $CN_D$ and $CN_S$ are the C:N ratio of decomposers and their substrates, respectively. Gray dots are the estimated microbial CUE of litter decomposition in natural terrestrial ecosystems from Manzoni *et al*. (2017). Black squares are the microbial CUE measured via laboratory incubation experiments of Gilmour & Gilmour, (1985), Devêvre & Horwáth (2000) and Thiet *et al*. (2006). Error bars represent the standard deviations. N min (g N $kg^{-1}$ soil) is the concentration of soil mineral N."

to

"**Figure 6** Comparison of $CUE_d$ (lines) predicted by Eq. (2) with parameter values ($m_2 = 0.54$, $n_1 = 0.50$) calibrated based on the incubation experiments (Table A2) of Recous et al. (1995) and Guenet et al. (2010)to observed $CUE$ of terrestrial microorganisms along a gradient of $CN_S/CN_D$. For observed CUE (dots), $CN_D$ and $CN_S$ are the C:N ratio of decomposers and their substrates, respectively. For simulated CUE (lines), $CN_S$ and $CN_D$ correspond to the C:N ratio of donor (litter pool) and acceptor (the active SOM pool of the CENTURY), respectively. Gray dots are the estimated microbial CUE of litter decomposition in natural terrestrial ecosystems from Manzoni *et al*. (2017). Black squares are the microbial CUE measured via laboratory incubation experiments of Gilmour and Gilmour, (1985), Dev êvre and Horw áth (2000) and Thiet *et al.*(2006). Error bars represent the standard deviations. $N_{min}$ (g N kg$^{-1}$ soil) is the concentration of soil mineral N." (lines 1494-1506)

**16. How is Fig. A2 different from Fig. 5? Moreover, the caption in A2 doesn't seem to match the display item? (see also lines 374, 376).**

The reviewer is correct: we have inserted a wrong figure as Fig. A2. Now we have corrected the error. Please see:

[Figure]

"

**Figure A3** Dynamic of the simulated carbon use efficiency (*CUE*) and *f(N$_{min}$)* during the incubation experiments (Table A4). *CN$_{lit}$* is the C:N ratio of incubated litter, and *N$_{min}$* is the initial soil mineral N concentration (g N kg$^{-1}$ soil). M0-M3 are the four models in Table 1. Here the simulation results of each model were calculated with parameters optimized based on all of the 14 samples of incubation experiments (Table A2)." (lines 1571-1577)

**17. From line 480-506 on the discussion wanders well beyond the scope of results presented here. In particular, the emphasis on humic substances and litterfall driving SOM formation seems well out of line with contemporary thinking about factors controlling SOM stabilization (Lehmann and Kleber 2015). Moreover, the positive connection between CUE and steady-state SOM pools in first order models is well established (e.g. Frey et al. 2013). What's nice with the work presented here is the ability to link ideas about litter quality and SOM formation in ways that are consistent with theory about CUE and substrate quality (MEMs conceptual model, Cortufo et al. 2013) in a first order model. I'd encourage the authors more closely stick to interpreting the results presented with this work.**

Thanks for your suggestion. We have revised the manuscript to make it more closely stick to interpreting the results presented with this work. The original sentences have been changed from "This study provides some insights on processes leading to increased SOM sequestration. Soil C sequestration plays a crucial role in food security and land CO$_2$ emission (Lal, 2004). The international initiative '4 per 1000' has been proposed to increase global SOM stock by 0.4% per year to compensate for anthropogenic CO$_2$ emissions (Baveye *et al.*, 2018). Transforming more plant litter into stable SOM (e.g. humic substances) has been suggested as an effective strategy to sequester more C in soil (Prescott, 2010). Our model results show a positive linear relationship between equilibrium SOC stock and CUE of decomposed litter (Fig. A4). This result can also be interpreted by calculating the analytical equilibrium SOC storage of a fully linear model including only one litter pool and one SOC pool. In such a model, SOC receives C from the litter at a rate $CUE_d \times D$, where $D$ is the litter decomposition rate, which equals to litterfall at steady state. SOC is lost via first order decay with a decay constant $k$. At steady state, input to and outputs from the SOC pool are equal and thus,

$$CUE_d \times D = k \times SOC \rightarrow SOC = CUE_d \frac{D}{k} \tag{11}$$

With a mean residence time of C in the SOC between 10 and 20 years and $D$ approximated by litterfall (Table A4), SOC at equilibrium is predicted to scale linearly with $CUE_d$, with a slope approximately between 20 and 40, consistent with results in Fig. A4.

Therefore, litter quality needs to be controlled to maximize C sequestration in SOM pool (Eq. (2)). In line with previous studies (Prescott, 2010; Smith, 2016), our model predicts that adding N through fertilization and N-fixing plants will not only increase litter decay but also the fraction of litter-C being transformed into SOM and ultimately SOC stocks. However, application of mineral N fertilizer is associated with risk not considered here, like increasing land $N_2O$ emission (Mosier and Kroeze, 2000; Kanter *et al*., 2016; Yi *et al*., 2017) and causing nitrate leaching which in turn can induce water pollution (Cao *et al*., 2006; Strokal *et al*., 2016). Due to the negative environmental impacts of mineral N addition, the use of N-rich litter substrates for increasing SOM is advised."

to

"This study provides insight on processes leading to increased SOM sequestration. Enhancing the efficiency at which plant residuals are transformed into stable SOM has been suggested as an effective strategy to sequester C in soil (Prescott, 2010; Cotrufo *et al*., 2013). Simulation results from our model suggest a positive linear relationship between equilibrium SOC stock and CUE of decomposed litter (Fig. A4), in line with the earlier findings with a similar model (for example Frey *et al*. 2013). In fact, with linear models such as CENTURY it can be shown that the steady state SOC scales linearly with CUE, different from nonlinear models predicting that higher CUE can trigger SOC loss (Allison *et al*., 2010). Our model goes beyond earlier attempts (Bonan *et al.*, 2013; Fujita *et al.*, 2014; Averill and Waring, 2018) by adapting the optimal metabolic regulation hypothesis of Manzoni *et al.* (2017) to link CUE, litter quality and SOM formation in a process-oriented way." (lines 729-740)

**18. Line 512, self-citations are nice, but it may also be worth referencing other modeling groups here?**

We have changed the original sentence from "An increasing number of land surface models (e.g. ORCHIDEE-CNP, Goll *et al.*, 2017) have representations of the terrestrial N cycle."

to

"An increasing number of land surface models (Wang *et al.*, 2010; Zaehle *et al.*, 2014; Goll *et al.*, 2017) have representations of the terrestrial N cycle." (lines 1035-1036)

**19. Line 516, didn't Bonan and others (2013, cited elsewhere in the text) already do this with CLM and CENTURY? Seems worth crediting work that's already been done along. these lines.**

Although the constraint of soil mineral N availability on litter decomposition rate has been represented in some land surface and soil biogeochemical model (Bonan *et al.*, 2013; Fujita *et al.*, 2014; Averill and Waring, 2018), to our knowledge, none of these models have tested the links CUE to litter stoichiometry and soil nutrient availability. However, we acknowledge that other theoretical models have included this link (Schimel and Weintraub, 2003). The original sentence in our manuscript might have not given an accurate statement. We thus changed it from "By incorporating our litter decomposition formulation in these land surface models that simulate the dynamics of soil mineral N concentration, it will be possible to test and validate our developments with more extensive data from laboratory and field experiments."

to

"By incorporating our newly developed formulations of $CUE_d$ and $f(N_{min})$ in these land surface models that simulate the dynamics of soil mineral N concentration, it will be possible to test and validate our developments with more extensive data from laboratory and field experiments." (lines 1036-1040)

**20. Line 538, the comparison with 'most large-scale litter decay models' was not made in this paper and I would remove this clause from the conclusion.**

Thanks for your suggestion. We have removed this clause. The original sentence is changed from "Overall, the developed model captures the microbial mechanisms mediating litter stoichiometry and soil mineral N effects on litter decomposition and SOM formation – representing an improvement over most existing large-scale litter decay models."

to

"Overall, the developed model captures the microbial mechanisms mediating litter stoichiometry and soil mineral N effects on litter decomposition and SOM formation." (lines 1059-1066)

**Response to Referee #2**

**1.This study adapted a conceptual formulation of CUEd based on assumption that litter decomposers optimally adjust their CUEd as a function of litter substrate C to nitrogen (N) stoichiometry. The new model algorithm was incorporated into CENTURY soil biogeochemical model and evaluated using data from laboratory litter incubation experiments. The results showed that new CUEd formulation with flexible CUE and effect of N availability to decay rate was able to reproduce differences in respiration rate of litter with contrasting C:N ratios and under different levels of mineral N availability. It is well-written, logically organized, and the figures and tables are appropriate.**
Thanks for your positive comments.

**2. Figure 1 seems too simple to include other major processes mentioned in the**

**method section. It should be considered to revise.**

Thanks for your reminding. We have revised the Fig. 1 and checked the Method section to make sure that all important processes have been illustrated in this flowchart. Finally, we added the temperature ($T$ (℃)) and soil moisture ($SWC$ (%)) factors for scaling litter decay rate, as well as the inhibition effect of mineral N on litter decay rate ($f(N_{min})$). The original Fig. 1 has been changed from

[Figure]

"

**Figure 1** Schematic diagram of the C flows in the litter decay model used in this study. $f_m$ is the fraction of metabolic compounds in plant litter. $D(C_{lit-met})$ and $D(C_{lit-str})$ are the decomposition rates (g C kg$^{-1}$ day$^{-1}$) of metabolic or structural litter, respectively. $LC_{lit}$ is the lignin:C ratio (on a mass basis) of plant litter; $CN_{met}$, $CN_{str}$, $CN_{act}$, and $CN_{slow}$ are the C:N ratio of metabolic litter pool, structural litter pool, active SOM pool and slow SOM pool, respectively; $N_{min}$ is the concentration of mineral N in solution (g N kg$^{-1}$ soil); $CUE_d$ is C use efficiency of the transformation from litter to soil organic matter (SOM); $f_{SA}$, $f_{SS}$ and $f_{SR}$ are the fractions of decomposed structural litter-C that is transferred to active SOM pool, slow SOM pool and released to atmosphere in forms of $CO_2$, respectively. As in the algorithms in CENTURY model (Parton *et al.*, 1988), here $f_{SA} = CUE_{d\_SA} \times (1-f_{lig})$, $f_{SS} = CUE_{d\_SS} \times f_{lig}$, $f_{SR} = 1-(f_{SA}+f_{SS})$, where $f_{lig}$ is the lignin fraction (0–1, dimensionless) in the structural litter pool, and $CUE_{d\_SA}$ and $CUE_{d\_SS}$ are the CUE of C transformation from structural litter pool to active and slow SOM pool, respectively."

to

[Figure]

"

**Figure 1** Schematic diagram of the C flows in the litter decay model used in this study. $f_m$ is the fraction of metabolic compounds in plant litter. $D(C_{lit\text{-}met})$ and $D(C_{lit\text{-}str})$ are the decomposition rates (g C kg$^{-1}$ day$^{-1}$) of metabolic or structural litter, respectively. $LC_{lit}$ is the lignin:C ratio (on a mass basis) of plant litter; $CN_{met}$, $CN_{str}$, $CN_{act}$, and $CN_{slow}$ are the C:N ratio of metabolic litter pool, structural litter pool, active SOM pool and slow SOM pool, respectively; $N_{min}$ is the concentration of mineral N in solution (g N kg$^{-1}$ soil); $f(N_{min})$ is a factor reducing litter decay rate when soil mineral N availability is limiting; $T$ (°C) and $SWC$ (%) are temperature and soil water content, respectively; $CUE_d$ is C use efficiency of the transformation from litter to soil organic matter (SOM); $CUE_{max}$=0.8 is the maximum microbial CUE (dimensionless) when growth is limited by C from the organic substrate; $f_{SA}$, $f_{SS}$ and $f_{SR}$ are the fractions of decomposed structural litter-C that is transferred to active SOM pool, slow SOM pool and released to atmosphere in forms of $CO_2$, respectively. As in the algorithms in CENTURY model (Parton et al., 1988), here $f_{SA}=CUE_{d\_SA}\times(1\text{-}f_{lig})$, $f_{SS}=CUE_{d\_SS}\times f_{lig}$, $f_{SR}=1\text{-}(f_{SA}+f_{SS})$, where $f_{lig}$ is the lignin fraction (0–1, dimensionless) in the structural litter pool, and $CUE_{d\_SA}$ and $CUE_{d\_SS}$ are the CUE of C transformation from structural litter pool to active and slow SOM pool, respectively." (lines 1428-1445)

**3. As the CUEd was defined as a fraction of it is respired to the atmosphere and the remaining fraction (Line 159-160), it is not correct to use 1-CUEd to simulate CO2 emission in Fig. 1.**

Microbial carbon use efficiency (CUE),defined as the ratio of microbial biomass production to material uptake from substrates (lines 68-69). In our study, the CUE of decayed litter-C ($D_{(C\text{-}lit)}$) is defined as the ratio of C that is transferred into SOC pool (CUE×$D_{(C\text{-}lit)}$)to the total decayed litter-C. Therefore, the remaining fraction ((1-CUE)×$D_{(C\text{-}lit)}$) is respired to the atmosphere as CO2. To explain the definition of CUE more explicitly, we have changed the original sentence from "When C is being transferred between pools, a fraction of it is respired to the atmosphere and the remaining fraction (CUE$_d$ conceptually equal to microbial CUE) enters the acceptor pool."

to

"When litter is being decomposed, a fraction of the decomposed C is respired to the atmosphere and the remaining fraction (CUE$_d$ conceptually equal to microbial CUE) enters the acceptor SOM pool." (lines 196-199)

**4. Equ (4) is important for this study, which has been used to develop one of model simulations (i.e. M1). However what is the fundamental assumption for adding N effects in the Equ (4)? N mineralization is accompanied with carbon decomposition. So, why use N availability to limit litter decay?**

Biomass of microbes is stoichiometrically constrained. When the supply of N from substrates is lower than the demand of microbes to fulfill their specific stoichiometric C:N ratio, microbes will utilize the mineral N (immobilization). Thus low availability of mineral N can limit microbial activity, and in turn litter decay rate. There is no explicit representation of microbial growth in CENTURY model. But the C:N ratio of SOM pool is assumed to be same to that of the microbial biomass. The mineralized N accompanying with litter decomposition will preferentially enter SOM pool to fulfill the SOM C:N ratio. When the N supply from decomposed litter is lower than the demand of newly formed SOM, soil mineral N will be immobilized. Therefore when soil mineral N concentration is very low and the immobilized N cannot meet the N demand of SOM, then the mineral N becomes a constraint factor of litter decomposition rate.

We have provided a brief introduction on the fundamental assumption for adding the mineral N factor in Eq. 4, and it can be find from:"Microbial biomass is nearly homeostatic (Cleveland and Liptzin, 2007; Franklin *et al.*, 2011; Allison, 2012).
When the supply of N from substrates is lower than the demand of microbes to fulfill
their specific stoichiometric C:N ratio, microbes will utilize mineral N
(immobilization) (Manzoni *et al.*, 2012). Thus, low availability of mineral N can
limit microbial activity, and in turn litter decay rate (Manzoni and Porporato 2009;
Fujita *et al.*, 2014)." (lines 130-135)

and

"The C:N ratio of SOM (around 9:1 on a mass basis in CENTURY) is representative
of the decomposer biomass, its value being between the average C:N ratio of soil
microbial communities including fungi and bacteria (7.4:1 in Cleveland and Liptzin,
2007) and the C:N ratio of soil fungi (13.4:1 in Zhang and Elser, 2017), which are
probably largely responsible for fresh litter decomposition." (lines 259-264)

**5. Line 71: need reference here.**

We have added references for our statement: "During litter decomposition, only a
part of the decomposed litter-C is being transferred into SOM, while the remaining C
is being released as $CO_2$ to the atmosphere by microbial respiration (Campbell and
Paustian, 2015; Cotrufo *et al*., 2015)." (lines 73-76)

**6. Line 212: typo "The The C:N ratio"**

Sorry for the mistake. We have changed the original sentence from "The The C:N
ratio of SOM (around 9:1 on a mass basis in CENTURY) is representative of the
decomposer biomass, its value being close to the observed average C:N ratio of soil
microbes (7.4:1 in Cleveland and Liptzin, 2007 and 13.4:1for soil fungi in Zhang *and*
Elser, 2017)."

to

"The C:N ratio of SOM (around 9:1 on a mass basis in CENTURY) is representative
of the decomposer biomass, its value being between the average C:N ratio of soil
microbial communities including fungi and bacteria (7.4:1 in Cleveland and Liptzin,
2007) and the C:N ratio of soil fungi (13.4:1 in Zhang and Elser, 2017), which are probably largely responsible for fresh litter decomposition." (lines 259-264)

[revised manuscript text omitted]

